# Glucosinolate structural diversity shapes recruitment of a metabolic network of leaf-associated bacteria

Kerstin Unger[1], Syed Ali Komail Raza[1], Teresa Mayer[1,5], Michael Reichelt [2], Johannes Stuttmann[3], Annika Hielscher [4], Ute Wittstock [4], Jonathan Gershenzon [2] & Matthew T. Agler [1] ✉

Host defenses can have broader ecological roles, but how they shape natural microbiome recruitment is poorly understood. Aliphatic glucosinolates (GLSs) are secondary defense metabolites in Brassicaceae plant leaves. Their genetically defined structure shapes interactions with pests in *Arabidopsis thaliana* leaves, and here we find that it also shapes bacterial recruitment. In model genotype Col-0, GLSs (mostly 4-methylsulfinylbutyl-GLS) have no clear effect on natural leaf bacterial recruitment. In a genotype from a wild population, however, GLSs (mostly allyl-GLS) enrich specific taxa, mostly Comamonadaceae and Oxalobacteraceae. Consistently, Comamonadaceae are also enriched in wild *A. thaliana*, and Oxalobacteraceae are enriched from wild plants on allyl-GLS as carbon source, but not on 4-methylsulfinylbutyl-GLS. Recruitment differences between GLS structures most likely arise from bacterial myrosinase specificity. Community recruitment is then defined by metabolic cross-feeding among bacteria. The link of genetically defined metabolites to recruitment could lead to new strategies to shape plant microbiome balance.

Plant health depends to a great extent on microbial colonization of roots and leaves[1]. Besides pathogens which are detrimental to plant health, other microbes also play important roles in plant fitness. Non-pathogenic bacteria, especially, are important for protecting plants against microorganisms that can cause disease. For example, non-pathogenic bacteria enable plant survival upon germination in soil in the presence of potentially detrimental soil fungi[2] and can protect leaves against pathogen attack[3,4]. Thus, it is important to understand which factors determine the colonization of bacteria in organs like leaves. In this context, both plant-microbe and microbe-microbe interactions are relevant. While we have a good understanding of 1-on-1 interactions between pathogens and plants in leaves, it is mostly unknown how non-pathogenic bacteria survive there, and in turn which host traits shape their assembly into communities in naturally colonized plants.

In order to successfully colonize leaves, all bacteria need to overcome several hurdles, such that the taxa that finally reach the surface and endosphere of plant leaves have been filtered by several factors[5,6]. First, to find their way onto or into the leaf, microbes must overcome physical hurdles such as low water availability[7] and regulated stomatal openings[8]. Next, they need to evade the plant immune system[9,10], which is made up of sensors of microbial molecular patterns (pattern-triggered immunity and effector-triggered immunity) as well as an arsenal of defensive secondary metabolites. Finally, the leaf environment is thought to be oligotrophic and very heterogeneous[11,12], making it a challenge to find nutrient sources.

[1]Institute for Microbiology, Plant Microbiosis Group, Friedrich Schiller University Jena, Jena, Germany. [2]Department of Biochemistry, Max-Planck-Institute for Chemical Ecology, Jena, Germany. [3]CEA, CNRS, BIAM, UMR7265, LEMiRE (Rhizosphère et Interactions sol-plante-microbiote), Aix Marseille University, 13115 Saint-Paul lez Durance, France. [4]Institute of Pharmaceutical Biology, Technische Universität Braunschweig, Braunschweig, Germany. [5]Present address: Schülerforschungszentrum Berchtesgaden, Didactics of Life Science, Technical University of Munich, Munich, Germany. ✉e-mail: matthew.agler@uni-jena.de

The plant immune system, especially, is thought to play important roles in selection and regulation of bacterial colonizers. For example, flagellin proteins of bacteria are finely tuned to evade pattern-triggered immunity[13], and generation of oxidative stress is important for regulating opportunistic pathogens[14]. On the other hand, little is known about how immune components shape colonization of non-pathogenic bacteria. Plant leaves produce a diversity of secondary metabolites, many of which are well-known for contributions to defense and are potentially toxic to a wide range of organisms. Here, we focused on the well-known glucosinolate-myrosinase system and asked how it might influence leaf bacterial communities of healthy *Arabidopsis thaliana* plants. Glucosinolates (GLSs) are secondary metabolites produced by plants in the Brassicaceae and related families. They share a common backbone structure consisting of a β-D-glucopyranose residue linked via a sulfur atom to a (Z)-N-hydroximinosulfate ester with variable side chains[15]. The chemical diversity of GLSs is determined by their side chains, which result from different amino acids and different modifications during GLS biosynthesis[16]. Aliphatic GLSs are a diverse group of GLSs derived from methionine, alanine, leucine, isoleucine or valine, whereas indole or benzenic GLSs are synthesized from aromatic amino acids[15]. In *A. thaliana*, the plant genotype defines the ability to synthesize a certain set of aliphatic GLSs, but the precise GLS mixtures are controlled developmentally, organ-specifically and in response to environmental factors. Wild genotypes isolated across Europe are typically characterized by a single major leaf aliphatic GLS that defines a "chemotype"[17].

Aliphatic GLSs are constitutively present particularly in epidermal cells and in specialized cells along the vascular bundles[18]. In addition, up to 5% of total leaf aliphatic GLSs may be present on the leaf surface[19]. Although considered biologically inactive, GLSs can be activated upon leaf damage by myrosinases, which hydrolyze the GLS, removing the glucose moiety and leading to rearrangement to various breakdown products, including isothiocyanates (ITCs), nitriles, and epithionitriles. The final chemical mixture depends on the aliphatic GLS structure and the presence of plant specifier proteins[20], as well as on abiotic conditions like temperature and pH[21]. A schematic overview of GLS hydrolysis is provided in Supplementary Fig. 1.

The role of aliphatic GLSs and their breakdown products have been best studied with respect to their defensive role against herbivorous insects. However, ITCs especially are well-known for antimicrobial properties against a broad range of plant and human pathogens in vitro[22,23] and in the model *A. thaliana* genotype Col-0, they help protect against bacterial and fungal pathogens[24,25]. In turn, microbial pathogens have adapted to the Brassicaceae with mechanisms to deal with toxic breakdown products such as detoxification and efflux pumps encoded by *sax* (survival in *Arabidopsis* extracts) genes to cope with ITC stress during infection[24,25]. While ITCs have long been considered to be present only after activation upon plant cell damage, there is now evidence for a constant turnover of GLSs to ITCs and cysteine as part of sulfur-cycling in plants[26]. Indeed, 4MSOB-ITC was detected in the apoplastic fluids of healthy Col-0 leaves[27] and the low concentrations present were reported to be enough to affect *P. syringae* virulence. Similarly, GLSs and their breakdown products exuded from roots are known to affect rhizosphere bacteria community assembly[28] and fumigation of soils with ITCs or bulk biomass from Brassicaceae plants suppresses detrimental microorganisms in soils[29,30]. Therefore, we hypothesize that aliphatic GLSs and their breakdown products might function as a filtering mechanism for bacterial leaf colonization of healthy plants. Given the wide chemical differences among aliphatic GLSs and their breakdown products in different plant genotypes, we further reasoned that different GLSs may shape the leaf bacterial community in distinct ways.

## Results

### Wild *A. thaliana* populations from Jena have distinct aliphatic GLS profiles

We studied five distinct, wild populations of *A. thaliana* located in Jena, Germany (Figs. 1A, C, Supplementary Tab. 1). We had previously isolated individual plants from these populations, grown them in the laboratory, and characterized their leaf GLS profiles[31]. The chemotype of all the isolates differed from that of the model genotype Col-0, where 4MSOB-GLS is the principal aliphatic GLS (Fig. 1B). The main GLS in three out of five isolates was 3-hydroxypropyl GLS (3OHP-GLS) (SW1, JT1, PB). In one isolate (NG2) allyl-GLS dominated, and another isolate (Woe) produced both 2-hydroxy-3-butenyl GLS (2OH3But-GLS) and allyl-GLS. In 2022 and 2023 we additionally analyzed the GLS profiles of NG2 and Woe plants sampled directly from wild populations. We found that Woe GLSs were the same as those previously extracted from this population, but wild NG2 contained both 2OH3But-GLS and allyl-GLS, similar to Woe (Supplementary Fig. 2). Since plants of all these Jena populations possessed a completely different aliphatic GLS composition than the widely used reference genotype Col-0, we compared one of them, NG2, to Col-0 to understand how GLS diversity affects the assembly of leaf bacterial communities.

### Aliphatic GLS breakdown products of certain *A. thaliana* genotypes inhibit growth of commensal leaf bacteria

We assumed that inhibition of bacterial growth would be the most likely mechanism by which aliphatic GLSs or their breakdown products would shape bacterial leaf communities. To compare toxicity between GLS-derived products in *A. thaliana* Col-0 and NG2, we homogenized leaves to mix the GLSs with myrosinases and release GLS breakdown products into the medium. We prepared what we refer to as "leaf extract medium" from the isolated genotype NG2 (which mainly produces allyl-GLS) and the reference genotype Col-0 (mainly 4MSOB-GLS), and from two transgenic lines, Col-0 *myb28* (with reduced aliphatic GLSs, Fig. 1B) and Col-0 *myb28/myb29* (with no aliphatic GLSs, Fig. 1B). We tested the leaf extract media against 100 diverse bacterial isolates recovered from *A. thaliana* leaves collected from the wild populations NG2 and PB (Supplementary Data 1, Supplementary Fig. 3). For most isolates, growth in Col-0 leaf extract medium was poor. Isolates of *Curtobacterium* spp., *Xanthomonas* spp. and *Pseudomonas* spp. all grew slightly better with aliphatic GLS-reduced *myb28* leaf medium, and all tested genera grew better with aliphatic GLS-free *myb28/myb29* leaf medium indicating growth inhibition by aliphatic GLS breakdown products. Interestingly, NG2 leaf extract medium was less inhibitory, and several strains even grew significantly more than in Col-0 *myb28/myb29* extract (Fig. 2A). These opposing effects suggest that aliphatic GLS breakdown products and potentially other leaf metabolites from different *A. thaliana* genotypes act very differently towards bacteria.

Assuming that ITCs would be the main inhibitory compounds in leaf extract medium, we tested pure ITCs against the bacterial isolates. As expected, 4MSOB-ITC inhibited growth of most isolates, especially gram-positive strains, consistent with the broad inhibitory effect of Col-0 leaf extract. Gammaproteobacteria like *Stenotrophomonas* sp., *Xanthomonas* and *Pseudomonas* spp. were more resistant (Fig. 2B). For the strains of the latter two genera this greater resistance corresponds to the presence of *sax* genes which are known ITC resistance genes and for *Pseudomonas syringae* strains to the ability to degrade 4MSOB-ITC (Supplementary Fig. 4, Supplementary Tab. 2). Allyl-ITC was also toxic, especially to gram-negative colonizers in apparent contradiction to the results with the NG2 leaf extract where these bacteria grew well (Fig. 2C). Upon investigation of the actual GLS breakdown products, we found that Col-0 leaf homogenates contained high levels of 4MSOB-ITC (29.6 ± 6.1%) and 4MSOB-CN (39.3 ± 7.4%), the corresponding nitrile. In NG2 homogenates, however, the predominant aliphatic GLS breakdown product was not the corresponding ITC, but

rather the epithionitrile 3,4-epithiobutanenitrile (CETP, 86.6 ± 2.1%), known to be derived from allyl-GLS by the action of additional plant specifier proteins[20]. Only a minor proportion of allyl-GLS was converted to the corresponding nitrile (allyl-CN, 5.4 ± 0.7%) and even less to allyl-ITC (0.3 ± 0.0%) (Fig. 2D, chemical structures in Supplementary Fig. 1B). Thus, this epithionitrile, although found in leaf homogenates at high levels, apparently hardly inhibited the growth of most bacterial isolates. Together, although NG2 and Col-0 likely differ in several ways chemically, the aliphatic GLS structure and the types of breakdown products influence toxicity towards leaf colonizing bacteria. This led us to hypothesize that these metabolites in particular would underlie important differences in colonization of non-pathogenic leaf bacteria in the two genotypes.

### Aliphatic GLSs do not decrease bacterial colonization, but rather increase colonization of specific taxa in the NG2 genotype

Based on the apparent higher toxicity of Col-0 leaf homogenates, we reasoned that there would be higher potential for GLSs to affect leaf bacterial community assembly in Col-0 than in NG2 plants. To test both genotypes together, we knocked out *myb28* in the NG2 background, which completely eliminated aliphatic GLSs in the leaves of this genotype (Fig. 1B). Then, we grew both genotypes and their respective aliphatic GLS-free mutants in natural soil collected in Jena and performed 16S rRNA gene amplicon sequencing to characterize the bacterial community in surface-sterilized (mostly endophytic bacteria) and whole (including all surface bacteria) leaves. Importantly, we used hamPCR[32] so that results are normalized to single-copy host gene abundance and reflect differences in absolute bacterial abundances, not just relative abundances.

We did not find any significant differences in alpha or beta diversity of endophytic or total leaf bacterial communities between Col-0 and its aliphatic GLS-free mutant *myb28/myb29* (Supplementary Data 2, Supplementary Fig. 5), agreeing with previous work[27]. In NG2, we also did not observe differences in endophytic communities (Supplementary Data 2, Supplementary Fig. 5), but the beta diversity of total leaf communities was significantly affected by the *myb28* knockout (Fig. 3B, PERMANOVA: $R^2 = 0.14704$ $p = 0.042$ Jaccard; $R^2 = 0.2472$ $p = 0.065$ Bray-Curtis, Supplementary Figs. 6,7). Because the difference between whole leaves and endophytes are the removal of leaf surface bacteria, this suggests that mainly leaf surface bacteria were affected, and only in NG2. To our surprise, NG2 also had *higher* total bacterial loads in leaves compared to NG*myb28* (Wilcoxon test, $p = 0.032$, Fig. 3A). Further, DESeq2 identified 14 differentially abundant taxa affected by genotype ($p < 0.05$ for 5 with the more stringent Wilcoxon rank-sum test) and all but one were *enriched* in NG2 compared to NG*myb28* (Fig. 3C). Seven of these 13 belonged to the order Burkholderiales (families Comamonadaceae, Oxalobacteracae and Methylophilaceae). The strong enrichment of these taxa caused an overall enrichment of Comamonadaceae and Methylophilaceae, as well as an overall enrichment of the order Burkholderiales (Supplementary Fig. 6). The other taxa enriched in NG2 WT leaves were diverse but included members of the Rhizobiales and the Flavobacteriales. Notably, these taxa were not aways the most abundant in the leaf samples (Supplementary Fig. 7). Other taxa, including Sphingomonadaceae and Pseudomonadaceae, reached higher abundance in some replicates, but were not consistent. Together, the results refuted our hypothesis that the toxicity of GLS breakdown products affected commensal leaf colonization of healthy plants. Instead, the enrichment of bacteria in NG2 with aliphatic GLSs compared to NG*myb28* suggests a strong positive effect of allyl-GLS, but not 4MSOB-GLS on specific taxa.

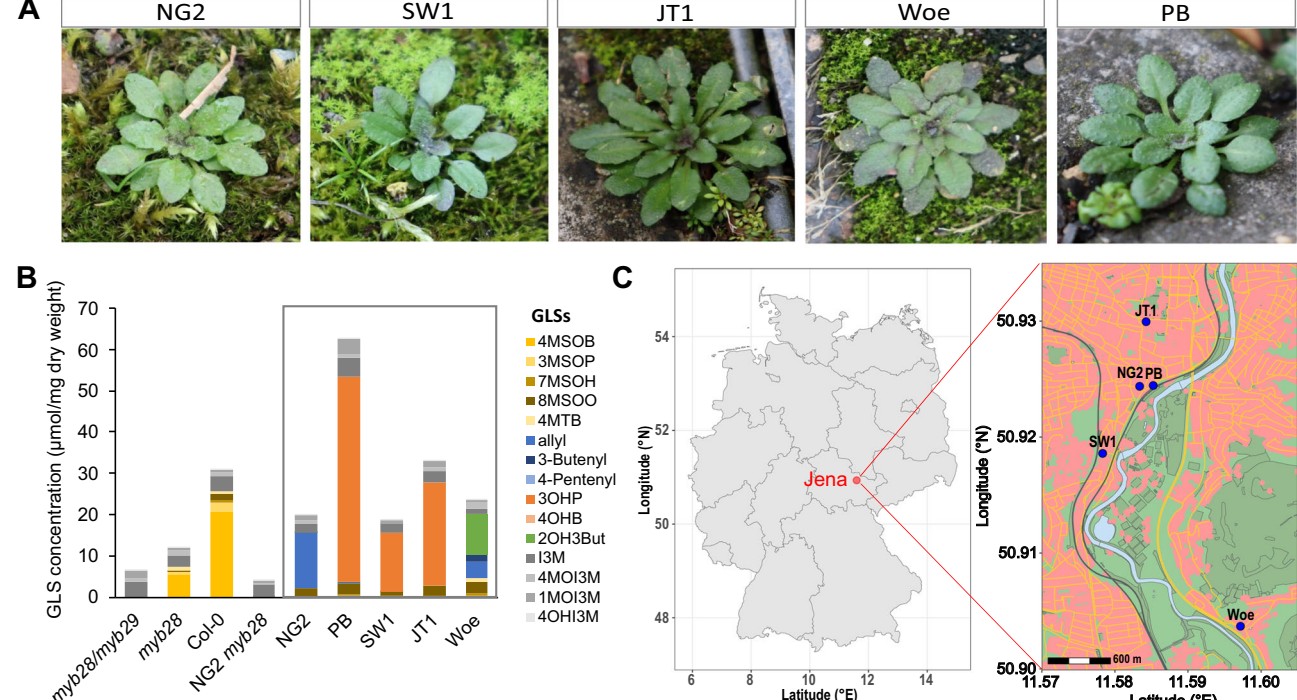

**Fig. 1 | Local *A. thaliana* populations in Jena produce distinct GLS profiles.** We used these differences to study the impact of these leaf metabolites on bacterial community composition. **A** Individual *A. thaliana* plants of the five selected populations in Jena, Germany: NG2 (Neugasse), SW (Sandweg), JT1 (Johannistor), Woe (Wöllnitz), PB (Paradiesbahnhof) in February 2022. **B** Average GLS concentrations of 3–4 replicate rosettes of the five local *A. thaliana* populations (gray box), the reference genotype Col-0 and respective aliphatic GLS mutants in Col-0 and NG2 background. Colored GLSs are aliphatic, gray shades are indole GLSs. Abbreviations for GLSs are listed in the methods. **C** Map of Germany and Jena showing the sampling locations of the five populations (generated using Open-StreetMap data) additional information is available in Supplementary Tab. 1. Data underlying these figures are available in our figshare folder (see Data Availability).

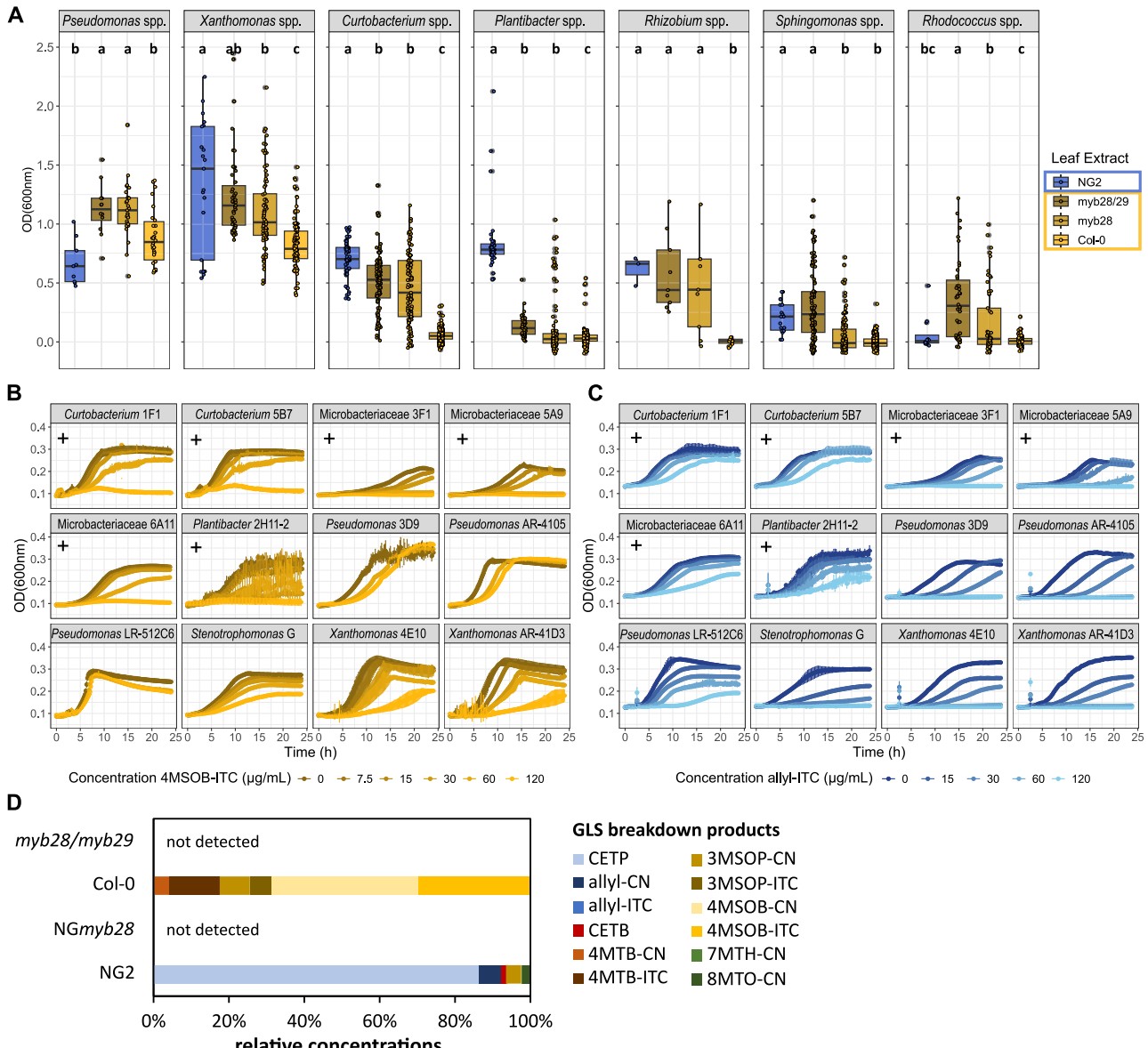

**Fig. 2 | Effects of GLS degradation products on growth of diverse leaf colonizing bacteria. A** Final $OD_{600}$ of bacterial strains grown in leaf extract media of *A. thaliana* ecotypes NG2, Col-0 and Col-0 mutants *myb28* and *myb28/myb29*. Each strain was measured in three technical replicates and at least in two leaf extract media. Some strains were tested repeatedly, and data of several strains was agglomerated at genus level for better visibility. Number of strains per genus: 7 *Pseudomonas*, 12 *Xanthomonas*, 16 *Curtobacterium*, 13 *Plantibacter*, 2 *Rhizobium*, 37 *Sphingomonas*, 13 *Rhodococcus* spp. Letters indicate statistical significance based on ANOVA followed by a Tukey post-hoc test with alpha = 0.05, two-sided. (see also: individual plots in Supplementary Fig. 3; individual n and results of the post-hoc tests in Supplementary Data 4). **B, C** Growth curves of a set of 12 bacterial strains in

*A. thaliana* leaves in R2A medium supplemented with 4MSOB-ITC (**B**) or allyl-ITC (**C**). Gram-positive strains are marked with a +, the remaining strains are gram-negative. Mean and standard deviation of three replicates are shown per condition. **D** Relative concentration of aliphatic GLS breakdown products in NG2, NG*myb28*, Col-0 and *myb28/myb29* leaf homogenates per gram fresh weight. The average of three replicate rosettes per genotype is shown. CN = nitrile, ITC = isothiocyanate; additional details on the abbreviations for GLS breakdown products are listed in the methods section. Chemical structures of the main breakdown products are provided in Supplementary Fig. 1B. Data underlying these figures are available in our figshare folder (see Data Availability).

## *A. thaliana* leaves in the wild enrich specific taxa associated with aliphatic GLSs

To test whether the bacterial taxa enriched in the lab *in planta* in response to aliphatic GLSs are also enriched in *A. thaliana* in the wild, we sampled leaves of *A. thaliana* together with leaves of other sympatric, ground-dwelling ruderal plants growing in our five wild populations (Fig. 1) and characterized whole leaf bacterial communities. Bacterial communities associated with *A. thaliana* leaf samples were similar to those of other plants, but not identical. 2.7% of the variation in community composition corresponded significantly to the plant

type (*A. thaliana* vs. other plants), 5.6% when considering the interaction of plant type with location (Fig. 4A). Differential abundance analysis did reveal an *A. thaliana*-specific signature involving several taxa, some of which were enriched in *A. thaliana* leaves both across all locations (Fig. 4B) and at the NG2 location (Supplementary Figs. 8,9). Among these, taxa in the order Burkholderiales stand out because they belonged to one of the same families (Comamonadaceae) and included some of the same genera (*Acidovorax, Rhodoferax*, and *Methylophilus*) as those enriched in NG2 vs. NG*myb28* in the lab experiments (Fig. 3C). *A. thaliana* in populations PB and SW1, both with a 3OHP-GLS

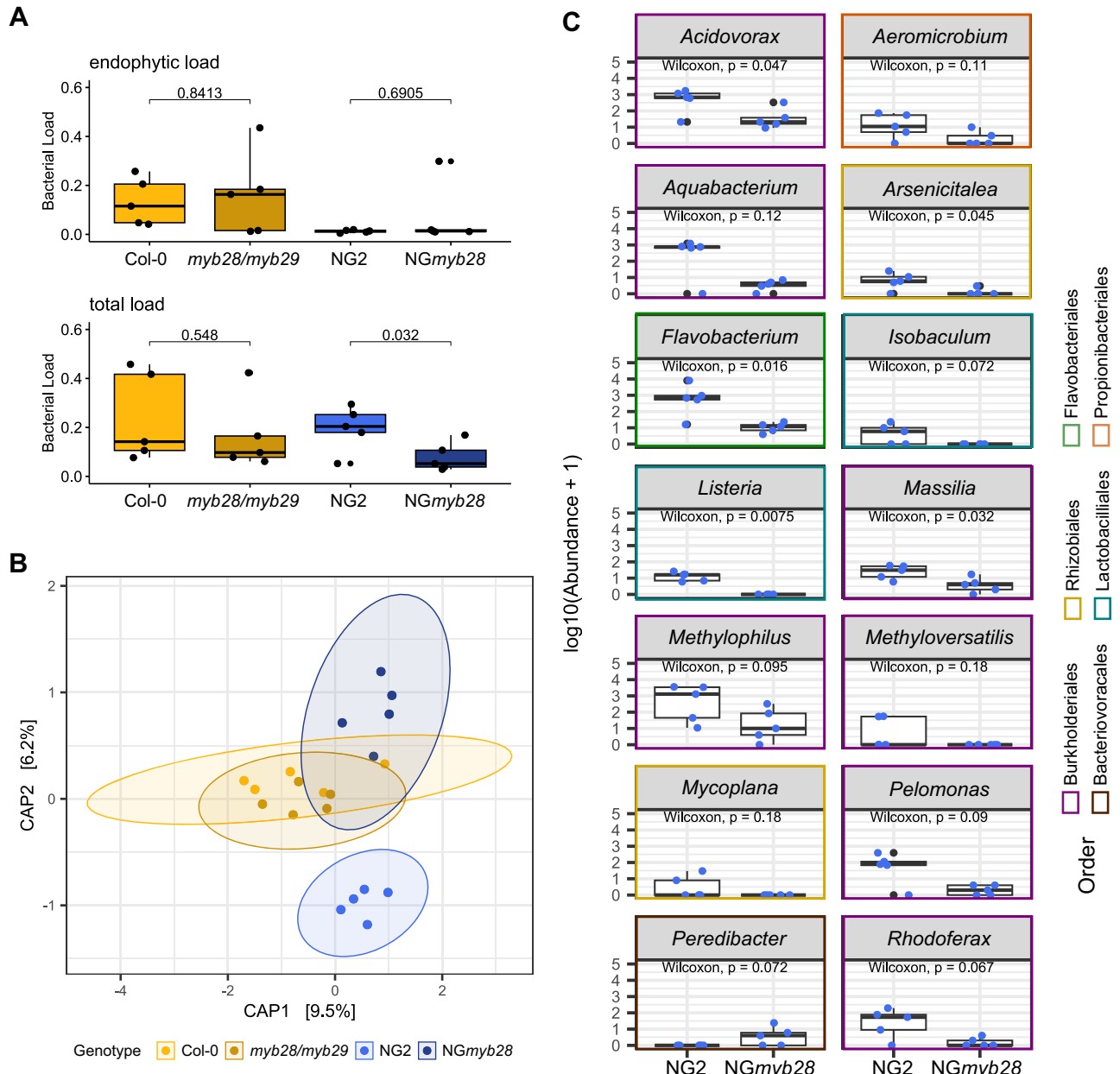

**Fig. 3 | Bacterial community analysis of leaves with and without aliphatic GLSs in NG2 and Col-0 background.** Bacterial community composition of whole rosettes of 3-week-old plants was assessed by amplicon sequencing of 16S rRNA genes ($n$ = 5). **A** Bacterial loads of total and endophytic leaf communities of *A. thaliana* assessed by normalization to plant GI reads, pairwise Wilcoxon test (two-sided, p was adjusted using fdr) was used to check for significances. Each dot represents one rosette. **B** Beta diversity of total leaf communities visualized as constrained PCoA of Jaccard index. Pairwise comparisons (PERMANOVA, 999 permutations, between NG2-NG*myb28*, Col-0-*myb28/myb29*) revealed significant differences between NG2-NG*myb28*. **C** Differentially abundant taxa on NG2 compared to NG*myb28* plants using DESeq2 analysis with a cutoff of alpha = 0.05. Significant taxa were log10-transformed and plotted, and a more stringent Wilcoxon test (two-sided, no p-adjustment) was used to provide additional information on the strength of the enrichment. The colors of the boxes show the order level. Data underlying these figures are available in our figshare folder (see Data Availability).

chemotype, as well as Woe, another allyl-GLS-containing genotype, also enriched Comamonadaceae genera, but overall, fewer taxa than NG2 (Fig. 1B, Supplementary Fig. 8). At higher taxonomic levels, only Methylophilaceae and Comamonadaceae were enriched at the NG2 location. Thus, specific genera within a few Burkholderiales families are strongly correlated with allyl-GLS in NG2 and some overlapping genera within the same families are also robustly enriched in *A. thaliana* compared to other plants that do not produce GLSs in the wild. Together with other enriched taxa, this indicates that allyl-GLS supports recruitment of specific bacteria to the leaves of *A. thaliana*. Differences between NG2 and Woe, with the same GLS chemotype

(Supplementary Figs. 2,8) may suggest that other genotypic or ecological factors shape recruitment locally.

## Enrichment of bacteria on allyl-GLS as the sole carbon source parallels *in planta* enrichments

We hypothesized that bacteria on the leaf surface of the NG2 *A. thaliana* population may utilize surface allyl-GLS as a carbon source, leading to the enrichment of certain strains and overall higher bacterial loads on NG2 plants compared to NG*myb28*. Supporting this hypothesis, we were able to detect the expected GLS in extracts generated by dipping individual leaves in methanol, confirming that leaf surface

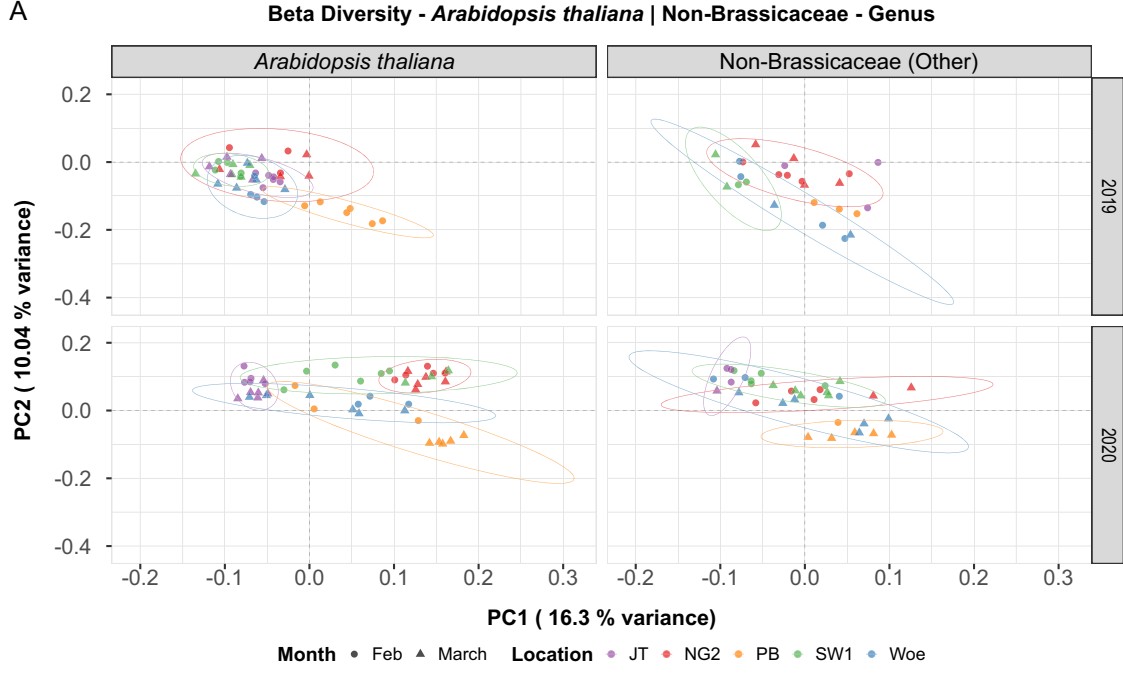

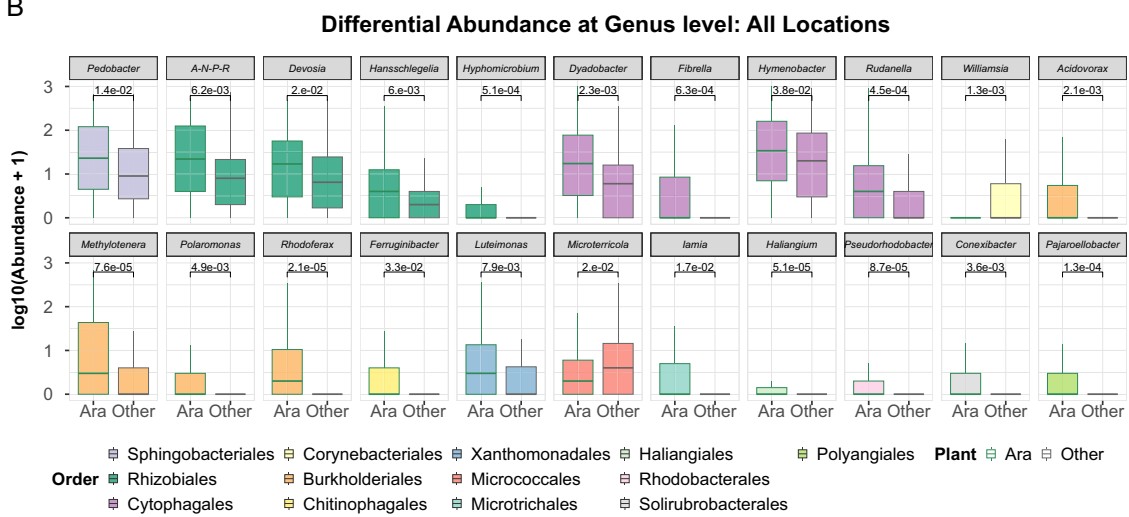

**Fig. 4 | Leaf bacterial community signature of *A. thaliana* compared to sympatric non-Brassicales plants across years and locations.** Leaf bacterial community compositions in five locations (NG2, JT1, PB, Woe, SW1) in February and March of 2019 and 2020 assessed by amplicon sequencing of 16S rRNA genes. **A** PCoA of leaf bacterial communities based on Aitchison distances from centered log ratio (CLR)-transformed genus-level compositions. Points represent individual samples, categorized by location (color) and month (shape). 95% confidence intervals are shown for each location-year combination via ellipses. PCoA was performed on all data together and the plots are facetted by year and plant type. Variance explained is indicated on the axes. The fraction of variance explained by

the factors are based on PERMANOVA (1000 permutations): Location = 17.2% ($p = 0.001$), Year = 8.5% ($p = 0.001$), Plant = 2.7% ($p = 0.001$), Month = 1.6% ($p = 0.001$), Location x Year = 9.1% ($p = 0.001$), Location x Plant = 5.6% ($p = 0.001$). **B** Differentially abundant taxa using DESeq2 analysis with a cutoff of alpha = 0.05. Additionally, a Wald test with a parametric fit was executed to determine significant differences. Only significant taxa were log10-transformed and plotted with *p*-values calculated using the Benjamini−Hochberg method (two-sided). Boxplots are colored by order. A-N-P-R = Allorhizobium-Neorhizobium-Pararhizobium-Rhizobium. Data underlying these figures are available in our figshare folder (see Data Availability).

bacteria should have access to GLSs (Supplementary Fig. 10). To test if bacteria can grow on them, we washed bacteria from leaves of wild NG2 plants to inoculate liquid M9 minimal medium supplemented with allyl-GLS, 4MSOB-GLS or glucose as the sole carbon source. Nitrogen and sulfur, also found in GLSs, were not limited in the base medium. An additional experimental trial followed one passage in glucose with two passages in allyl-GLS (Fig. 5A). The leaf surface wash contained $5.43 \times 10^6$ CFU/mL. We enriched for three passages, where each time 10% of the volume was transferred to a new substrate so that any remaining leaf carbon sources would have been insignificant

(~1000× diluted). Passage intervals were adjusted to the time until the medium became visibly turbid in the first passage. This was seven days for allyl-GLS, 14 days for 4MSOB-GLS, three days for glucose, and five days for glucose followed by allyl-GLS. After the final passage, CFUs were counted. Bacterial populations reached an average of $3.26 \times 10^9$ CFU/mL on allyl-GLS-supplemented medium, $1.80 \times 10^8$ CFU/mL on 4MSOB-GLS, and $1.79 \times 10^9$ CFU/mL on glucose followed by allyl-GLS (Fig. 5A).

16S rRNA gene amplicon sequencing showed that the final communities grown on glucose- or 4MSOB-GLS-supplemented medium

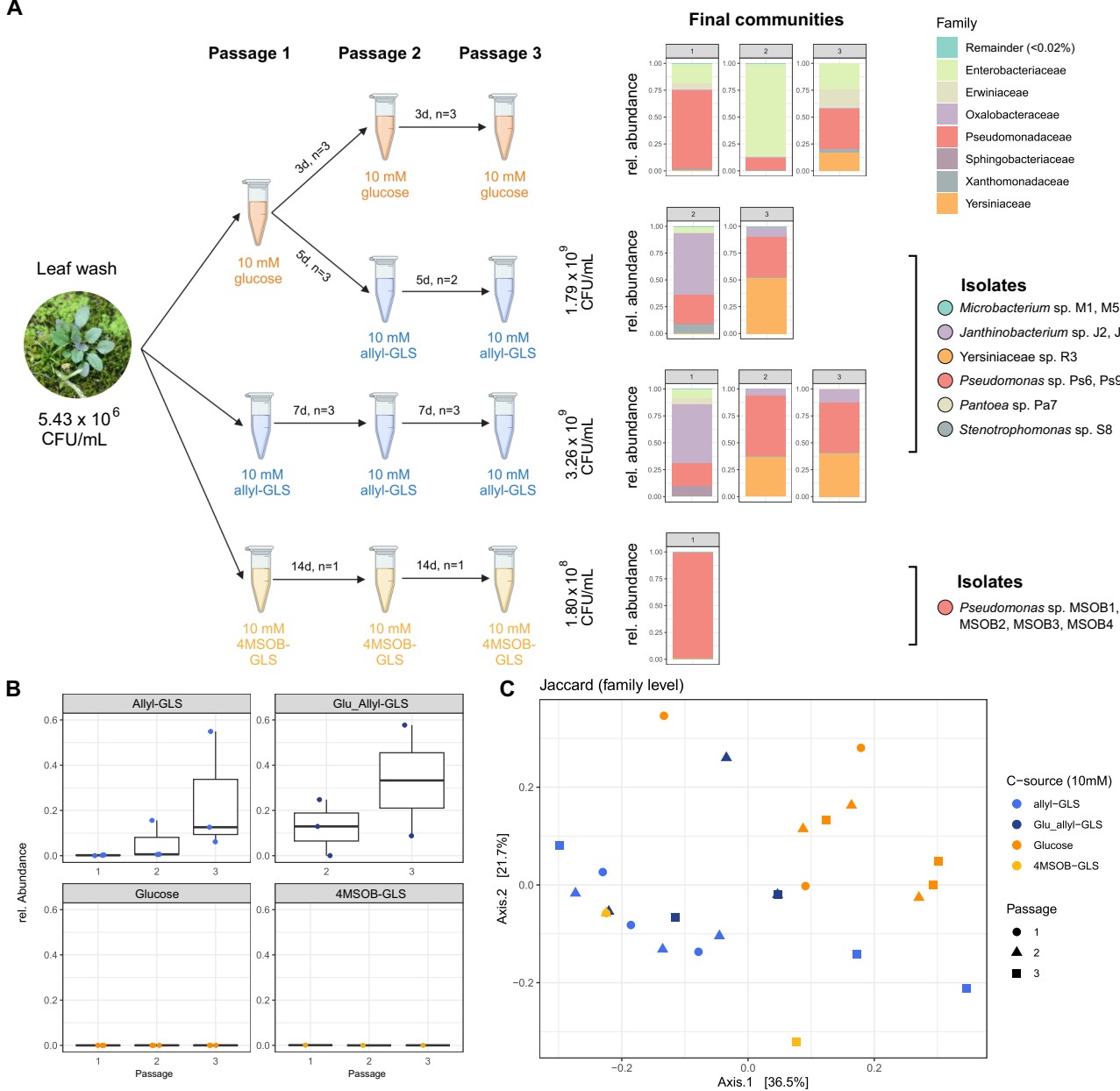

**Fig. 5 | Enrichment of bacterial strains from NG2 leaf surface on different aliphatic GLSs as sole carbon source. A** Scheme of enrichment process (created with Biorender.com) with initial and final CFUs, growth intervals and number of technical replicates are given in the figure. M9 medium with different GLSs was incubated in tubes without shaking. Bar charts show the community composition after the third passage assessed by 16S rRNA gene amplicon sequencing; results for all other passages are shown in Supplementary Fig. 11. The charts show data agglomerated on family level. Families below 0.02% relative abundance were merged and classified as "Remainder". Only replicates with >100 reads were considered. The legend on the right side shows which strains were isolated from allyl-GLS and 4MSOB-GLS enrichments after the final passages. **B** Relative abundance of Oxalobacteraceae family over the three passages in all enrichments, n is the same as in the schematic drawing in (**A**). **C** Beta diversity measured by Jaccard distances of all enrichments on family level with significant differences based on C-source (PERMANOVA, 999 permutations: $p = 0.006$, $R^2 = 0.257$). Data underlying these figures are available in our figshare folder (see Data Availability).

were dominated by Pseudomonadaceae. On 4MSOB-GLS Pseudomonadaceae made up almost 100% of the total, whereas both Enterobacteriaceae and Pseudomonadaceae were abundant on glucose. The communities growing on allyl-GLS were distinct and showed one of two different configurations, each with at least one member of the order Enterobacterales, one Pseudomonadaceae and notably, one Burkholderiales (always a *Janthinobacterium* ASV, belonging to Oxalobacteraceae). A similar community structure assembled on allyl-GLS from a divergent community that had been pre-enriched on glucose (Supplementary Fig. 11). In three of five replicates, one Yersiniaceae ASV (Enterobacterales) dominated together with Pseudomonadaceae

and Oxalobacteraceae (Fig. 5A). In the other two replicates, a community developed that was dominated by Oxalobacteraceae and Pseudomonadaceae but also included Enterobacteriaceae and Erwiniaceae (both Enterobacterales). ASVs belonging to Oxalobacteraceae increased in relative abundance over the course of the passages only when allyl-GLS was the carbon source (even after one passage on glucose, Supplementary Fig. 11). Thus, community formation was dynamic, with the Oxalobacteraceae increasing in importance over time only in allyl-GLS enrichments (Fig. 5B). In general, the carbon source was correlated to 25.7% of the variation between communities ($p = 0.006$, PERMANOVA, Jaccard) (Fig. 5C). In conclusion, when the

carbon source was allyl-GLS, specific communities developed. These were enriched in *Janthinobacterium*, an Oxalobacteraceae, reminiscent of the robust enrichment of bacterial genera within a few closely related Burkholderiales families *in planta* both in lab-grown plants and in wild populations.

## Only a Yersiniaceae strain metabolized aliphatic GLSs, with rates depending on the structure

We next isolated bacteria from the communities growing on medium with 4MSOB-GLS and allyl-GLS as sole carbon sources to determine which individual taxa directly utilize GLSs. The recovered taxa closely reflected those identified by amplicon sequencing (Supplementary Tab. 3, Fig. 5A). From the 4MSOB-GLS medium, four *Pseudomonas* isolates were recovered, but none grew successfully on 4MSOB-GLS within six days (Supplementary Fig. 12A). This makes sense given the 14-day growth time required to reach a relatively low cell density in the passages in 4MSOB-GLS medium, and underscores that bacteria grow only very slowly on 4MSOB-GLS. From the allyl-GLS medium, seven isolates representing most of the taxa detected at family level were tested, but only one Yersiniaceae strain (*Rahnella* sp., hereafter R3) grew on allyl-GLS within six days (Fig. 6A, Supplementary Fig. 12B). R3 metabolized an average of 69.1% of the allyl-GLS in the medium in six days of growth. Metabolite analysis of the culture supernatant showed that very little of the allyl-GLS was recovered as allyl-ITC ($0.006 \pm 0.001$ mM), but that on average 93.1% was metabolized to the presumably less toxic breakdown product allyl-amine ($11.6 \pm 4.7$ mM) (Fig. 6B). Fitting to this activity, we found that the R3 genome has close homologs of known plant- and soil-associated bacterial myrosinases with secretion signals (up to 75% positive amino acid match), as well as two homologs (up to 95% positive amino acid match) of functionally characterized ITC hydrolases (SaxA) (details in Supplementary Data 3).

To understand the specificity of the enrichment, we tested whether R3 could grow on 2OH3But-GLS, which wild NG2 plants produce in leaves together with allyl-GLS (Supplementary Figs. 3B,D), or on 4MSOB-GLS, produced by Col-0. Growth was observed on 2OH3But-GLS, but with a longer lag-phase (Fig. 6A)−only 20.9% of 2OH3But-GLS was consumed within nine days. In contrast to allyl-GLS, little of the 2OH3But-GLS was converted to the corresponding amine ($0.03 \pm 0.01$ mM), but instead ~58.9% was found as goitrin ($1.77 \pm 0.33$ mM) which results from spontaneous cyclization of the unstable 2OH3But-ITC (Fig. 6B). R3 did not grow on 4MSOB-GLS, agreeing with no observed enrichment of bacteria in Col-0 vs. Col-0 *myb28/myb29*. We reasoned that the slow/no growth patterns on 2OH3But-GLS and 4MSOB-GLS might be because of toxicity due to slow/no conversion of intermediate goitrin/4MSOB-ITC. However, when grown in the presence of the intermediates, R3 metabolized both 4MSOB-ITC and allyl-ITC to the corresponding amines (Fig. 6D). Even though it did not convert goitrin, none of the three GLS breakdown products strongly inhibited growth (Fig. 6C). Since these results reject a toxicity hypothesis, we reason that specificity at the level of GLS hydrolysis (possibly myrosinase specificity) currently best explains substrate specificity, and can also explain enrichment specificity *in planta*.

## Cross-feeding interactions support community growth on allyl-GLS

The previous results suggested that when allyl-GLS is the carbon source, only GLS-utilizing bacteria like R3 would be directly enriched, but we observed more diverse communities. One possibility is that bacteria that do not directly use GLSs rather grow on plant-derived GLS breakdown products, like CETP, that could be generated spontaneously or due to leaf damage. However, we consider this unlikely because among 13 tested bacteria, only a *Xanthomonas* strain grew better in NG2 leaf extract medium compared to NG*myb28* extract

medium, while some grew less, including an isolate enriched with R3 (Supplementary Fig. 13). Another possibility is that a synergistic effect occurs and that isolates that do not utilize allyl-GLS on their own do so in a community. To test this, we grew combinations of isolates that were abundant in the enrichment (*Janthinobacterium* J4, *Pseudomonas* Ps9 and *Stenotrophomonas* S8) on allyl-GLS with or without R3. However, we only observed growth when R3 was present (Fig. 7A).

A third possibility is that enrichment could be indirect if the growth of R3 on allyl-GLS as a carbon source supports the growth of other taxa via metabolite cross-feeding. Supporting this, R3 with *Pseudomonas* Ps6 or Ps9 reached a higher maximum $OD_{600}$ than R3 alone, suggesting more efficient carbon utilization (Supplementary Fig. 12). Ps9 also grew quickly in R3 spent medium, produced from R3 grown on allyl-GLS as sole carbon source (Fig. 7B, C). J4, on the other hand, did not grow in the same spent medium. To investigate dynamics in a longer-term co-culture similar to the enrichment process shown in Fig. 5, we grew synthetic communities of combinations of Ps9, J4 and R3 (representing the major taxa in one of the enrichment conformations) and passaged them three times in M9 medium with allyl-GLS as the sole carbon source, then characterized the bacterial communities and supernatant metabolomes. The experiment was repeated twice in two different volumes with similar community characterization results. In all communities after three passages, R3 and Ps9 dominated, while J4 also persisted at lower levels (Fig. 7D, Supplementary Fig. 14). This closely reflects the conformation in the allyl-GLS enrichments when Yersiniaceae and Pseudomonadaceae dominated. Untargeted metabolomic analysis of the supernatant from the third passage showed that in all cases, >90% of allyl-GLS was utilized. A small feature corresponding to 3-Butenyl-GLS (apparently an impurity in the allyl-GLS standard used as C source), was similarly utilized. The metabolome in the supernatant of R3 alone was significantly altered in the presence of other bacteria (Fig. 7E). Multiple features apparently produced by R3 when it grew alone, including adenine and guanine, were reduced by >90% in R3 + Ps9 and the addition of J4 resulted in further decreases of mostly the same features (Fig. 7F). Together, these results strongly suggest that growth on allyl-GLS by a single bacterium can support a more diverse community as part of bacterial metabolic networks. Interestingly, we also observed that residual allyl-ITC was progressively lower in communities with more partners (down to ~50 μM) compared to R3 alone (~225 μM) (Fig. 7G).

## Discussion

Plant exudates are well-described to shape assembly of root and rhizosphere microbial communities[33,34] by serving as nutrient sources for rhizosphere bacteria[35], inhibiting growth of certain taxa to protect the plant[36,37] or altering microbial physiology and activity[38] and microbial interactions[39]. In leaves, however, few compounds are definitively known to positively recruit specific bacteria: Lab experiments have suggested that sugars non-specifically recruit leaf bacteria[40], while in the wild, positive recruitment of methylotrophic bacteria is known based on simple carbon compounds like methanol[41,42] that are thought to be by-products of plant metabolic processes[11]. Our findings show that recruitment in leaves may be even more prevalent and that secondary plant metabolites such as aliphatic GLSs can be important in orchestrating leaf colonization of commensal bacteria (Fig. 8).

Given the well-known antimicrobial effects of aliphatic GLS breakdown products[22–25], it may seem surprising that they did not negatively impact the colonization of *A. thaliana* leaves by commensal bacteria. However, previous work similarly found no suppressive effect of aliphatic GLSs on non-pathogenic bacteria in healthy Col-0[27]. In our work, some aliphatic GLSs were even found to promote the recruitment of specific bacterial taxa. The leaf surface is a challenging, oligotrophic environment with low availability of nutrients and water. Carbon sources, for example are known to be highly patchily

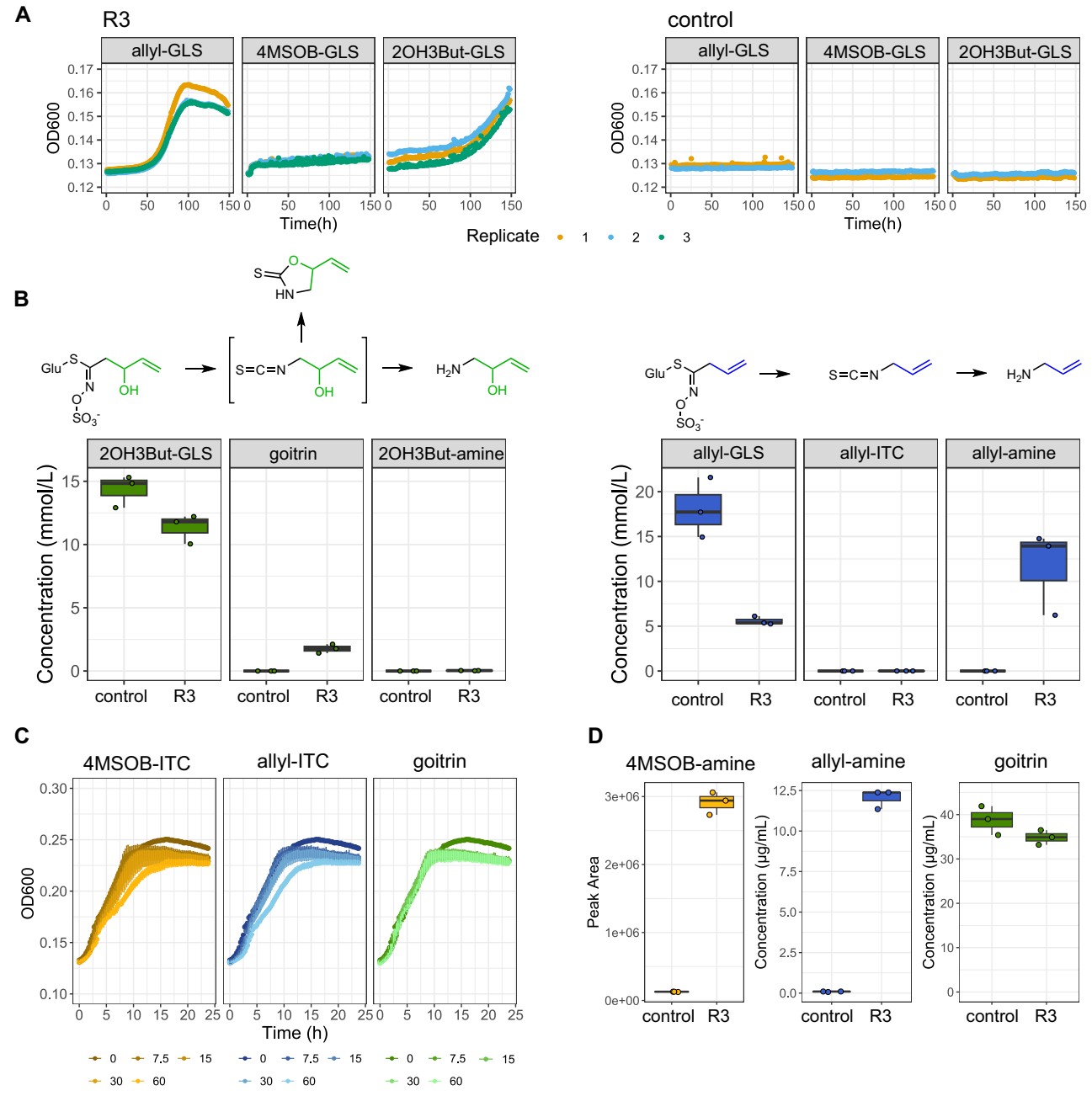

**Fig. 6 | Growth on and utilization of diverse aliphatic GLSs by R3. A** R3 growth in M9 medium supplemented with 10 mM allyl-GLS, 4MSOB-GLS or 2OH3But-GLS. $OD_{600}$ was measured every hour, inoculation with water served as negative control n = 3. **B** Analysis of 2OH3But-GLS, allyl-GLS and degradation products in R3 inoculated medium after six days (allyl-GLS) and nine days (2OH3But-GLS) of incubation. Each dot represents one replicate bacterial culture, n = 3. **C** Growth curves of R3 in R2A broth supplemented with 0–60 μg/mL 4MSOB-ITC, allyl-ITC or goitrin. $OD_{600}$ was measured every 15 min, mean and standard deviations of three replicates are depicted. The same three samples with DSMO instead of ITCs or goitrin served as control (0 μg/mL) in all three plots. **D** Quantification of 4MSOB-amine, allyl-amine and goitrin in 24 h-old cultures of R3 in R2A broth supplemented with 30 μg/mL of each of the ITCs or goitrin (n = 3). Data underlying these figures are available in our figshare folder (see Data Availability).

distributed[12]. However, plant secondary metabolites that can potentially be metabolized by leaf bacteria are present[11]. Aliphatic GLSs were previously found to be present on the leaf surface[19], with GLS biosynthesis genes found to be expressed at least in trichome cells[43]. Additionally, other non-defensive roles of GLSs are known. For example, adult cabbage butterflies (*Pieris rapae*) express gustatory receptors in their tarsi that sense allyl-GLS to identify host leaves for oviposition[44]. From a bacterial perspective, GLSs contain a glucose moiety, which can be enzymatically cleaved by myrosinases to yield

glucose. In addition, this hydrolysis reaction releases sulfate (S), and the further metabolism of breakdown products could provide more carbon or nitrogen in the form of amines. Supporting nutritive roles, human gut bacteria break down GLSs[45], and myrosinase-producing bacteria have been identified in both soil and plant roots[46,47] as well as in the phyllosphere[48]. Utilizing GLSs, however, results in formation of ITCs and other toxic products. Thus, resistance mechanisms to ITCs and other GLS breakdown products are also necessary for efficient GLS utilization.

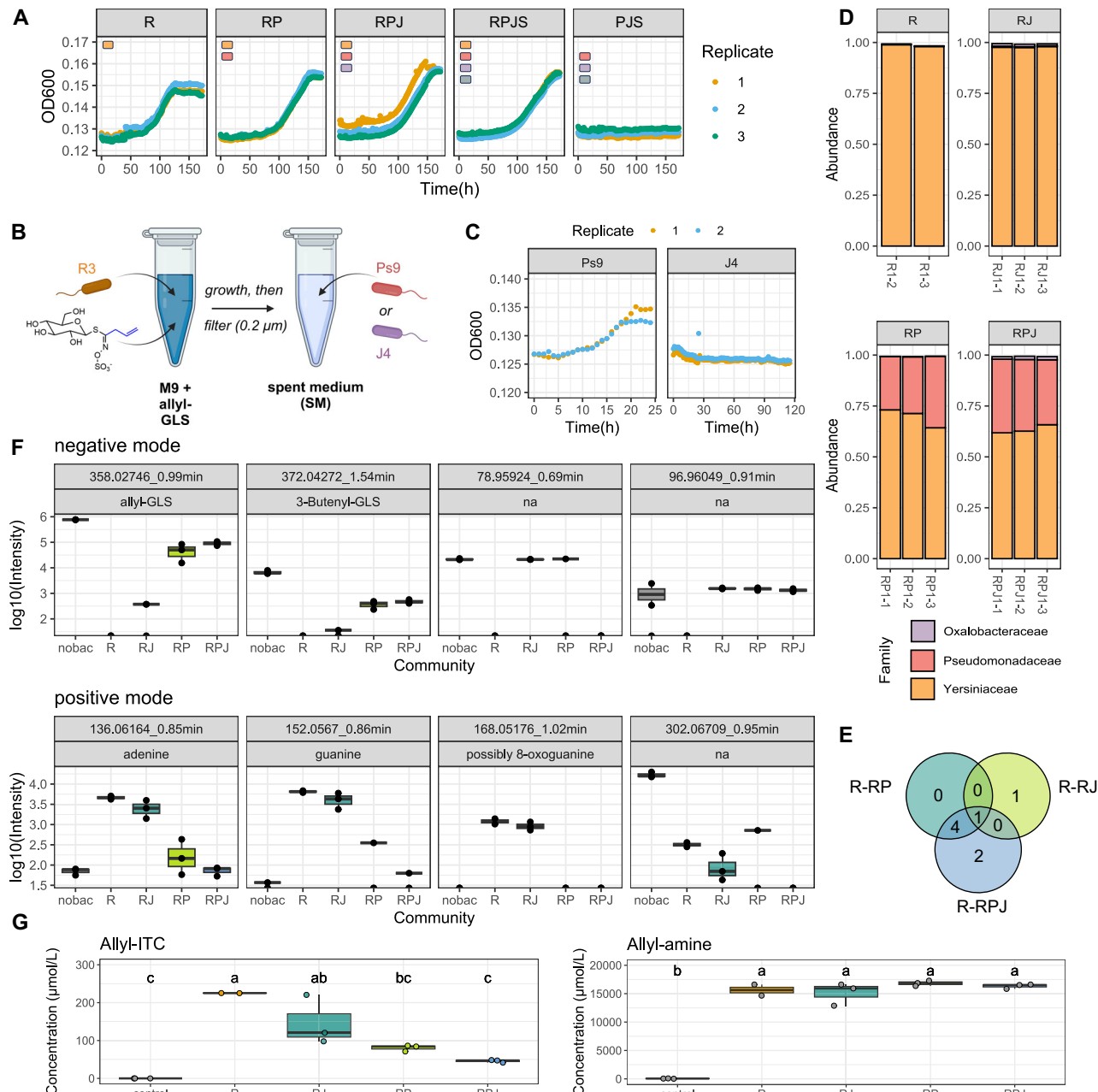

**Fig. 7 | Cross-feeding interactions based on R3 support community growth on allyl-GLS. A** Growth of combinations of isolates that were abundant in the enriched communities (*Janthinobacterium* J4, *Pseudomonas* Ps9 and *Stenotrophomonas* S8) in M9 medium with 10 mM allyl-GLS as the sole carbon source, with or without R3. **B** Scheme of experiment of J4 and Ps9 growth in spent media (SM) which was generated by growing R3 in M9 medium + allyl-GLS as carbon source. The figure was created with Biorender.com. **C** Growth of Ps9 or J4 in spent medium of R3. **D–G** Results of replicated experiment consisting of three consecutive passages of R3 (R), R3 + J4 (RJ), R3 + Ps9 (RP) and R3 + Ps9 + J4 (RPJ) in M9 medium with allyl-GLS as the sole carbon source (*n* = 3 for all combinations but R: *n* = 2). **D** 16S rRNA gene amplicon sequencing showing bacterial communities of the replicates. **E** Venn

diagram of significantly different metabolites in co-cultures compared to R3 mono-culture. **F** Non-targeted metabolomics of the culture supernatants, showing metabolites that had differential feature intensities ($p < 0.05$, corrected two-sided t-test, p-value fdr adjusted) when comparing the R3 culture with any of the R3 + other isolates. nobac = non-inoculated control. Peak statistics are provided in Supplementary Data 5. **G** Same as (**F**), but targeted metabolomics to quantify GLS break-down products allyl-ITC and allyl-amine. Letters indicate statistical significance based on ANOVA followed by a Tukey post-hoc test with alpha = 0.05, two-sided. Details on the statistical analysis are provided in Supplementary Data 6. Additionally, all data underlying these figures are available in our figshare folder (see Data Availability).

Here, we recovered one strain, *Rahnella* sp. R3, that was able to grow in minimal medium with aliphatic GLSs as the only carbon source, and testing it allowed us to build a model to explain the structural specificity of GLS bacterial recruitment (Fig. 8). R3 utilized allyl-GLS, but not the structurally divergent 4MSOB-GLS, which aligns with the fact that aliphatic GLSs in Col-0 did not apparently recruit leaf bacteria. We first presumed this was because the breakdown product

4MSOB-ITC prevented R3 from growing on it, because ITCs can be very toxic and affect activity and growth[27,49]. However, R3 grew in the presence of 4MSOB-ITC. Additionally, the R3 genome carries homologs of ITC hydrolases, which convert ITCs to non-toxic amines[24], and indeed could detoxify 4MSOB-ITC. Thus, a toxicity explanation seems unlikely, and we instead hypothesize that specificity arises at the myrosinase step. Supporting this, bacterial myrosinases are known to

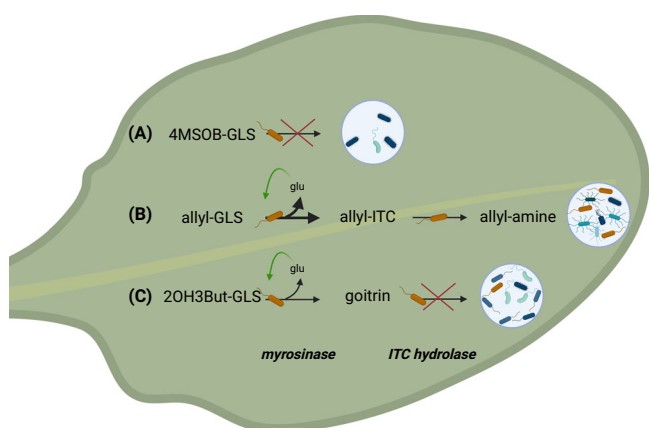

**Fig. 8 | Hypothesis for mechanisms of community assembly on GLS structural variants.** Our work suggests a scenario in which the specificity of GLS enrichment of bacteria arises at the level of the hydrolysis of GLSs by a myrosinase enzyme: **A** Bacteria do not grow on 4MSOB-GLS, but the corresponding ITC can be detoxified, suggesting that bacterial myrosinases either do not convert this GLS or are not expressed in its presence. We cannot exclude that it is metabolized slowly by some bacteria or other microorganisms, but this should result in relatively little bacterial enrichments. **B** Allyl-GLS is efficiently metabolized (thick arrow) because myrosinases hydrolyze it to allyl-ITC, which is in turn rapidly detoxified to allyl-amine. Availability of glucose drives primary growth, and other taxa cross-feed, which could involve allyl-amine but certainly involves other secreted metabolic byproducts. Other community members may provide positive feedback to improve community function, for example by limiting accumulation of ITCs (**C**) 2OH3But-GLS is slowly metabolized (thin arrow) and goitrin is not detoxified. Limited toxicity of goitrin to R3 suggests that either this GLS is slowly hydrolyzed by bacterial myrosinase, or that goitrin accumulation negatively feeds back to slow hydrolysis. Slower metabolism will likely reduce flow of metabolites to other community members, and thus reduce enrichment. The figure was created with Biorender.com.

show substrate specificity[46,50]. R3 also grew on 2OH3But-GLS, but more slowly, again possibly because of myrosinase substrate specificity. It is also possible, however, that buildup of goitrin, which results from spontaneous ring formation of the unstable 2OH3But-ITC[51,52], negatively feeds back to slow the GLS degradation (Fig. 8C).

One inconsistency in our results was that in in vitro enrichments we only found R3, a Yersiniaceae bacterium, that can metabolize GLS. However, in natural *A. thaliana* populations and in planta in the lab we did not observe high abundances or consistent enrichment of Yersiniaceae. A likely explanation is that other bacteria may be able to functionally replace Yersiniaceae *in planta*, preventing its consistent enrichment in the phyllosphere. Supporting this, many bacteria have been found that can utilize GLSs (including other Enterobacterales)[47]. It was also previously reported that myrosinase homologs were enriched in Brassicaceae phyllosphere and rhizosphere metagenomes compared to soil and that these genes belonged to not only Enterobacteriaceae, but also Firmicutes[48]. If so, it is likely that our enrichment conditions, which are different from the leaf environment, selected for only some GLS-utilizing bacteria. Additionally, it is possible that bacteria that can access GLSs in leaves do not need to be present in high abundances so that we simply did not detect Yersiniaceae. At any rate, it will be interesting for future work to look into the diversity of GLS utilizers to understand the broader ecological relevance of their utilization.

Although we only definitively identified one bacterial strain, R3, that could grow on allyl-GLS, it was enriched together with a substantially more diverse community. Co-enrichment may have had benefits on the overall community, since the most diverse synthetic communities had the lowest residual allyl-ITC levels. In our toxicity assays, levels in this range were found to be very inhibitory to many leaf bacteria. This co-enrichment could be explained via metabolic cross-feeding of taxa that could not themselves metabolize GLS. It is

likely that cross-feeding would also be relevant *in planta*, since it can support diverse communities on small numbers of primary metabolites[53] and we previously found that leaf bacteria, including commensal *Pseudomonas*, are exceptional cross-feeders[54]. In addition, GLS metabolism would result in an abundance of breakdown products like glucose or allyl-amine that may be valuable carbon or nitrogen resources for co-occurring taxa.

A remarkably consistent effect that we observed (in both lab and wild plants as well as in the in vitro enrichment) was enrichment of specific Burkholderiales bacterial taxa, mostly in the families Oxalobacteraceae and Comamonadaceae. Bacteria in these groups are common leaf colonizers where they can be important for plant health, linked to growth promotion[55] and antifungal properties[56]. In *A. thaliana*, they contribute to suppression of pathogenic fungi, which is required for survival when seedlings germinate in soil[2]. Therefore, understanding their enrichment may lead to ways to promote plant health via microbiomes. These same taxa are likely *in planta* to contribute further functions to the leaf bacteriome. In different wild *A. thaliana* populations, we previously identified a Comamonadaceae genus as a "hub", highly positively correlated to the abundance of other bacteria in leaves of wild *A. thaliana*[57] and this was later confirmed using abundance-weighted networks. This strongly suggests these bacteria are at the center of a tightly connected group of leaf-associated bacteria that increase and decrease in abundance together[58]. We hypothesize that this is because they play key roles in leaf metabolic networks, at least in part including GLS-based carbon economies. This is supported by work showing that Burkholderiales tend to be metabolically complex and are often observed as part of consortia degrading complex compounds[59]. In enrichments of bacteria from the surface of peppers grown on capsaicin as the sole carbon and nitrogen source, only a Comamonadaceae bacterium (*Variovorax*) with a *Pseudomonas* strain could grow, apparently via reciprocal exchange of capsaicin-derived carbon and nitrogen[60]. Additionally, some Burkholderiales have the capacity to fix nitrogen in the phyllosphere[61,62]. Such functions would be likely to contribute to overall bacterial growth on surface GLSs and probably on other leaf-derived resources, helping to explain why so much diversity was correlated to GLSs *in planta*. Future work should be aimed at dissecting these networks to better understand the resource economies in leaf communities.

Aliphatic GLSs and their breakdown products are some of the best-studied leaf secondary metabolites involved in defense against pathogens. For example, 4MSOB-ITC protects against non-host pathogens by direct antimicrobial effects[25] and by suppressing expression of the type III secretion system, a major virulence factor of the pathogen *Pseudomonas syringae*[27]. Diverse other ITCs also are known to inhibit plant and human pathogens[22,23]. This has likely led to selective pressure resulting in pathogen specialization for GLS-containing plants: *Pseudomonas syringae* DC3000, *Pectobacterium* spp. and the fungal pathogen *Sclerotinia sclerotiorum* all express *sax* genes which enable virulence in the presence of ITCs in *A. thaliana* and cabbage plants[24,25,63]. Accordingly, we found *sax* gene homologs in genomes of the opportunistic pathogens *Pseudomonas* and *Xanthomonas*, even though they were isolated from healthy leaves. *Pseudomonas* 3D9 and AR-4105 possess the ITC hydrolase SaxA and were able to degrade 4MSOB-ITC. On the other hand, most leaf-colonizing bacterial taxa were strongly inhibited by GLS breakdown products like 4MSOB-ITC in Col-0. Thus, even though ITCs did not shape colonization of healthy leaves as we and others observed[27], they would be released in large amounts during herbivory or due to necrotic pathogens, probably strongly re-shaping leaf bacterial communities. That might explain why herbivory increased *P. syringae* bacterial loads in leaves of *Cardamine cordifolia*[64], since *P. syringae* are likely to have *sax*-gene mediated tolerance to ITCs. Additionally, defense responses including the release of ITCs might also occur in apparently healthy

tissues due to small-scale responses[61] to local attack by opportunistic pathogens found in these leaves[65]. On the other hand, events leading to GLS breakdown would likely have less effect in NG2 leaves, where allyl-GLS breaks down to a presumably less toxic epithionitrile. Therefore, understanding how interactions between leaf-damaging organisms and microbiomes together affect plant fitness[64] will require both focusing in on localized, small-scale effects[66] and looking beyond the model *A. thaliana* genotype Col-0, into diverse other chemotypes.

There are several important directions for future work to enable possible applications of these findings. For one, it is necessary to evaluate how particular GLSs and/or GLS mixtures shape recruitment in nature. GLSs clearly play dual roles in recruitment and defense and our results suggest these roles will probably vary depending on leaf bacteria in a community context. Thus, to fully understand their roles in nature, it will be necessary to evaluate how effects in controlled lab-conditions are shaped by factors like both the biotic and abiotic environment. We were also so far unable to measure allyl-GLS consumption directly in leaves because there are still significant uncertainties about how to feed leaf bacteria *in planta* with specific metabolites in a controlled but realistic way. For example, it is unclear how surface GLS become available to bacteria and how realistic leaf localization can be artificially reproduced. However, metabolite localization together with feeding and tracing experiments would contribute to a better understanding of GLS-mediated community assembly by revealing GLS turnover rates and helping elucidate how breakdown products shape activity of the leaf bacterial community. At any rate, the finding that assembly of communities on aliphatic GLSs is plant chemotype-specific and thus defined by plant genomes sets the stage for developing new approaches to shape and maintain balance in plant leaf microbiomes.

## Methods

### Local *A. thaliana* populations in Jena
We worked with the widely used reference *A. thaliana* Col-0, its single and double knock-out mutants *myb28* and *myb28/myb29*[67] and the local genotype NG2. The latter was identified and isolated in spring 2018 as one out of five wild *A. thaliana* populations in Jena, Germany: NG2, PB, SW1, JT1, and Woe[31]. One individual plant of each population was propagated in the lab from a single seed for two generations to generate relatively uniform, homozygous lines for further experiments. Local Jena *A. thaliana* genotypes NG2, JT1, Woe and SW are already deposited in NASC database (N2110865- N2110868) and PB and NG*myb28* mutant will follow and are available upon request (Supplementary Tab. 1).

### Knock-out of MYB28 in local NG2 *A. thaliana*
To study effects of aliphatic GLSs in NG2 we generated an aliphatic GLS-free mutant in NG2 background by knocking out the MYB28 transcription factor using a genome editing procedure by an RNA-guided SpCas9 nuclease[68]. The plasmid pDGE347 was programmed for six target sites within MYB28 (AT5G61420; AAAAAACGTTTGATGG AACAGGG; TTCAAATTCTCATCGACCGTAGG; GATCGGGAGTATTGCT TGTCGG; GCTTCTAGTTCCAACCCTACGG; GAAACCATGTTGCAACT GGATGG; GAAACGTTTCTTGCAACTCAAGG). The respective plasmid (pDGE816) was transformed into *Agrobacterium tumefaciens* strain GV3101 pMP90 and plants of accession NG2 were transformed by floral dipping as previously described[69]. Floral dipping resulted in CRISPR-guided transformation events already in the germ cells of the plant and therefore $T_1$ generation seeds were screened for successfully transformed seeds (indicated by RFP expression in seeds)[70]. Primary transformants and non-transgenic individuals from the $T_2$ population were PCR screened and Sanger sequenced to isolate homozygous *myb28* lines using oligonucleotides myb28_2315F and myb28_2316R (Supplementary Table 4). Leaves of plants of $T_3$ or $T_4$ generation were used for GLS analysis to confirm the decrease in aliphatic GLS levels.

### Growth of plant material
All plants were grown in a climatic chamber (PolyKlima, Freising, Germany) at 18 °C/22 °C, 10 h/14 h, night/day with 75% light intensity. For propagation, seed production and GLS analysis the plants were sown on regular potting soil (4 L Florador Anzucht soil, 2 L Perligran Premium, 25 g Subtral fertilizer and 2 L tab water) and for amplicon sequencing of naturally colonized leaves, seeds were sown on sieved garden soil from Jena mixed with half the volume of perlite (Perligran Premium). For plant propagation or GLS analyzes, unsterilized seeds were sown directly and then vernalized for at least two days at 4 °C in the dark. For other experiments, seeds were first surface sterilized using 70% ethanol, followed by 2% bleach and washed 3x with sterile MiliQ water, then were vernalized in 0.1% agarose before sowing. In all cases the plants were thinned out two weeks after germination and if required they were later pricked into individual pots.

### Bacterial growth assays in leaf extract medium
Details on bacterial isolates and how they were recovered from wild plants are found in the supplementary methods and Data S1. We produced plant-based media ("leaf extract medium") according to[25] with minor changes: Leaves of 6-week-old plants (Col-0, *myb28*, *myb28/myb29*, NG2) were crushed with a metal pestle in R2A broth (1 mL broth for 1 g leaf fresh weight). Supernatants were recovered by centrifugation at maximum speed. Next, the leaf extract media were filter sterilized (0.2 μm) and frozen in aliquots at −80 °C. Bacterial isolates were pre-cultured for 24–96 h (depending on time required to reach an $OD_{600} \geq 0.2$). A 96-well flat-bottom plate was filled with 45 μL leaf extract per well and inoculated with 5 μL of normalized cultures ($OD_{600} = 0.2$) in triplicates. R2A broth (yeast extract 0.5 g/L, peptone 0.5 g/L, casein hydrolysate 0.5 g/L, glucose 0.5 g/L, soluble starch 0.5 g/L, $K_2HPO_4$ 0.3 g/L, $MgSO_4$ 0.024 g/L, sodium pyruvate 0.3 g/L, pH = 7.2 ± 0.2[71]) was added to the negative growth controls. The plate was incubated with constant shaking at 150 rpm at 30 °C. The incubation time reflected the time of the corresponding pre-culture (24–96 h). Directly before the final $OD_{600}$ measurement (VERSAmax™ Microplate Reader, MolecularDevices with SoftMax® Pro Software) 50 μL R2A broth + 0.02% Silwet L-77 was added to each well. Later experiments included NG*myb28* leaf extract medium and were performed the same way but measured on a TECAN Infinite M Plex plate reader with multiple readings per well. We used the average $OD_{600}$ of these multiple readings of each technical replicate for plotting. Raw data from the plate reader was analyzed with custom R scripts. Since growth behavior of strains of the same genus was usually similar, we agglomerated the data on genus level for plotting.

### Evaluation of ITC toxicity on bacterial growth
L-Sulforaphane (4-methyl sulfinyl butyl isothiocyanate, 4MSOB-ITC; ≥ 95%, CAS 142825-10-3, Sigma-Aldrich) or allyl-ITC (allyl isothiocyanate, AITC; 95%, CAS 57-06-7, Sigma-Aldrich) were dissolved in DMSO. 3 μL of one ITC or a DMSO control was added to 87 μL R2A medium in 96-well plates resulting in final concentrations ranging from 7.5 to 120 μg/mL. We added 10 μL of each culture normalized to $OD_{600} = 0.2$ to triplicate wells and covered the plate with a transparent plastic foil to prevent evaporation of the ITCs. The plate was incubated at 30 °C in a TECAN Infinite M Plex plate reader and the $OD_{600}$ was measured every 15 min after 1 min of orbital shaking and recorded using the software i-control 2.0. The raw data was processed with custom scripts in R.

### Bacterial growth assays on various aliphatic GLSs as carbon sources
Pre-cultures of individual isolates were washed 2× with 1 mL M9 medium without a carbon source and resuspended in the same medium. The $OD_{600}$ was normalized to 0.2 or 0.3. We dissolved 4MSOB-GLS

(glucoraphanin; ≥ 95%, CAS 142825-10-3, Sigma-Aldrich or Phytoplan Heidelberg, Germany), allyl-GLS (sinigrin; >95%, CAS 57-06-7, Phytoplan, Heidelberg, Germany) or 2OH3But-GLS (Progoitrin, >97%, CAS 21087-77-4, Phytoplan, Heidelberg, Germany) in sterile MiliQ water to produce 100 mM stocks. M9 minimal medium (12.8 g/L $Na_2HPO_4$, 3.1 g/L $KH_2PO_4$, 0.5 g/L NaCl, 1.0 g/L $NH_4Cl$, 0.5 g/L $MgSO_4$)[46] with final concentrations of 10 mM carbon source or MiliQ water as control was used. Experiments were performed in 96-well plates and the plate was covered with a transparent plastic foil during the incubation time to prevent evaporation. The plate was incubated at room temperature in a TECAN Infinite M Plex plate reader and the $OD_{600}$ was measured every hour after 1 min of orbital shaking. For the passaging experiment of synthetic communities, pre-cultures were normalized to OD = 0.8 and mixed with their respective partner strains or M9 medium without C-source to reach a final OD of 0.2 of each strain in the final inocula. The inocula were added to media 1:10 so that the starting $OD_{600}$ was approximately 0.02. Plates with SynComs in M9 broth supplemented with 10 mM allyl-GLS were incubated at room temperature with constant shaking at 200 rpm for 7 days in the first two passages, followed by an incubation in the plate reader as described above during the last passage (5 days). After each passage 10 μL of the previous passage were used as inoculum for 90 μL fresh M9 medium with 10 mM allyl-GLS. The raw data was processed with custom scripts in R.

## Enrichment of GLS-utilizing leaf colonizers

To recover isolates which can utilize allyl-GLS as sole carbon source we performed an enrichment similar to[46]. First, we produced a leaf wash by collecting 10 leaves of 5 NG2 *A. thaliana* plants from the wild NG2 population in spring 2023. For this, leaves were collected and combined in a 1.5 mL tube, they were stored on ice and brought back in the lab. 500 μL 10 mM $MgCl_2$ + 0.02% Silwet L-77 were added, and the tube was vortexed at lowest possible speed for 20 min. Next, we let the tube stand for ~5 min to settle down particles and 300 μL of the supernatant was transferred in a new tube. Bacterial load was determined in the leaf wash by plating a 10-fold dilution series on R2A agar (Carl Roth, Germany). The pH of M9 medium (1.28 g $Na_2HPO_4$, 0.31 g $KH_2PO_4$, 0.05 g NaCl, 0.1 g $NH_4Cl$) was adjusted to 7.0 and the medium was autoclaved for 15 min at 121 °C. $MgSO_4$ stock (5.0 g/L) was prepared separately and filter sterilized. For each enrichment, one 1.5 mL tube was filled with 70 μL liquid M9 medium, 10 μL $MgSO_4$, 10 μL GLS or glucose (100 mM stock) or MiliQ water as control and 10 μL inoculum. Each passage consisted of three replicates with C-source and microbial inoculum and one replicate with water instead of bacteria as control for sterility of the medium. 4MSOB-GLS passages only consisted of one replicate and no water control, due to limited availability of 4MSOB-GLS. Controls with water instead of C-source were inoculated with leaf wash in the beginning, but no growth was observed. The first passage was inoculated with 10 μL of the leaf wash, later passages were inoculated with 10 μL of the previous passage. The tubes were incubated at room temperature in the dark with no agitation. The incubation time and hence the interval of the passages depended on the time it took the first passage to become visibly turbid (between 3 to 14 days). After each passage, 70 μL were frozen at −80 °C with 30 μL 86% sterile glycerol to preserve the microbial communities. A 10-fold dilution series of the replicates of the last passages of allyl-GLS and 4MSOB-GLS enrichments were plated on R2A agar and incubated at 30 °C for 48 h. CFUs with different morphologies were picked and isolated. Bacterial strains were identified by sequencing the 16S rRNA gene region with 8 F/1492 R primers (sequences see Supplementary Tab. 4).

## Analysis of GLSs

GLS profiles were measured in leaves and in bacterial cultures. Leaves were collected from lab-grown 4-week-old individuals of our wild isolates from the Jena populations, from 3-week-old plants of $T_4$ generation of NG*myb28* plants to confirm the loss-of-function mutation and

wild NG2 and Woe plants. Each genotype was tested at least in three replicates. The plants were harvested by removing roots and flower stems as close to the rosette as possible. The whole rosettes were frozen in liquid nitrogen and kept at −80 °C until further processing. GLSs were extracted from freeze-dried leaf material with methanol and desulfo-(ds)-GLSs were quantified by HPLC coupled to a photodiode array detector as in ref. 72. For bacterial cultures, supernatants were recovered by removal of cells after centrifugation at max. speed. Intact GLSs in bacterial cultures were measured directly in the supernatant using HPLC-MS/MS. Details of the analysis procedures are found in the Supplementary methods. The following GLSs were detected in the samples: 3-hydroxypropyl GLS (3OHP), 4-hydroxybutyl GLS (4OHB), 3-methylsulfinylpropyl GLS (3MSOP), 4-methylsulfinylbutyl GLS (4MSOB), 2-propenyl GLS (allyl), S-2-hydroxy-3-butenyl GLS (S2OH3But), R-2-hydroxy-3-butenyl GLS (R2OH3But), 3-butenyl GLS (3-Butenyl), 4-pentenyl GLS (4-Pentenyl), 4-hydroxy-indol-3-ylmethyl GLS (4OHI3M), 4-methylthiobutyl GLS (4MTB), 6-methylsulfinylhexyl GLS (6MSOH), 7-methylsulfinylheptyl GLS (7MSOH), 8-methylsulfinyloctyl GLS (8MSOO), indol-3-ylmethyl GLS (I3M), 4-methoxy-indol-3-ylmethyl GLS (4MOI3M), and 1-methoxy-indol-3-ylmethyl GLS (1MOI3M). Results are given as μmol per g dry weight.

## Analysis of GLS breakdown products

GLS breakdown products were measured in leaf homogenates and in bacterial culture supernatants. To analyze GLS breakdown products in leaf homogenates, 120 to 300 mg fresh weight per sample of 6-week-old rosettes of NG2, NG*myb28*, Col-0 and *myb28/myb29* were harvested. 100 μL MES buffer (50 mM, pH = 6) was added for 100 mg leaf material, next the leaves were grinded with a clean metal pestle and the pestle was rinsed with additional 100 μL MES buffer into the sample. GLS breakdown products were extracted using dichloromethane and measured on a GC-MS and GC-FID. Detected breakdown products were: allyl cyanide (allyl-CN), allyl isothiocyanate (allyl-ITC), 1-cyano-2,3-epithiopropane (CETP), 1-cyano-3,4-epithiobutane (CETB), 4-methylthiobutyl cyanide (4MTB-CN), 4-methylthiobutyl isothiocyanate (4MTB-ITC), 3-methylsulfinylpropyl cyanide (3MSOP-CN), 3-methylsulfinylpropyl isothiocyanate (3MSOP-ITC), 4-methylsulfinylbutyl cyanide (4MSOB-CN), 4-methylsulfinylbutyl isothiocyanate (4MSOB-ITC), 7-methylthioheptyl cyanide (7MTH-CN), 8-methylthiooctyl cyanide (8MTO-CN). Amines in bacterial culture supernatants were measured in the aqueous medium by HPLC-MS/MS. Allyl-ITC and goitrin were extracted from the supernatants with dichloromethane and analyzed by GC-FID. Details of the analysis procedures are found in the Supplementary methods.

## Non-target metabolite analysis by LC-ESI-Q-ToF-MS

For non-target analysis of R3 mono-cultures compared to co-cultures of R3 with Ps9 and/or J4, ultra-high-performance liquid chromatography−electrospray ionization− high resolution mass spectrometry (UHPLC−ESI−HRMS) was performed with a Dionex Ultimate 3000 series UHPLC (Thermo Scientific) and a Bruker timsToF mass spectrometer (Bruker Daltonik, Bremen, Germany) as described in[73]. Details are further described in Supplementary methods.

## 16S rRNA gene amplicon sequencing

The approaches used for amplicon sequencing were slightly different depending on the dataset, an overview is provided in Supplementary Tab. 5. The four datasets are referred to specifically below.

**Material and leaf sampling.** To analyze bacterial communities in the enrichments in minimal medium (dataset 1, see Supplementary Tab. 5), the glycerol stocks collected after the passages (see above) were directly used for DNA extraction. To analyze the bacterial community of lab-grown NG2, NG*myb28*, Col-0, *myb28/myb29* leaves (dataset 2, see Supplementary Table 5), 4−6 rosettes of 3-week-old plants per

genotype were washed twice with 1 mL sterile MiliQ water by inverting the tube three times to collect "whole" leaves. To collect "endophytes", an additional 4–6 rosettess were surface-sterilized with 70% ethanol and 2% bleach (each 1 mL in 1.5 mL tube, 3× inverting) and washed twice with sterile MilliQ water afterwards. From 2019 to 2020, we collected *A. thaliana* leaf samples from the five different locations in Jena (Supplementary Tab. 1, Supplementary Supplementary Tab. 5, dataset 3). At the same time, a similar number of other neighbouring plants were sampled. Sampling was conducted during the early days of spring in February and March each year. For smaller *A. thaliana* plants, we sampled half of the rosette, while for larger ones, 2–3 leaves were collected. For other plants a similar amount of plant material was selected. The leaf material was washed with sterile MiliQ water three times and samples were brought back to the lab on ice. Plant material was frozen in screw cap tubes with two metal beads and ~0.2 g glass beads (0.25–0.5 mm diameter) each at −80 °C until further processing.

**DNA extraction.** Bacterial DNA from GLS enrichments (dataset 1) was extracted from glycerol stocks using an SDS buffer lysis protocol with RNAse A and Proteinase K treatments and a phenol/chloroform cleanup, followed by DNA precipitation. To extract DNA of lab-grown (dataset 2) and wild plants (dataset 3), a CTAB buffer and bead beating protocol was followed, with either a phenol-chloroform cleanup followed by precipitation (dataset 2) or magnetic beads (dataset 3). Precise details of each of the three protocols can be found in the Supplementary methods.

**Library preparation for amplicon sequencing.** In all library preparations ZymoBIOMICS Microbial Community DNA Standard II (ZymoResearch, Freiburg, Germany; referred to as "ZymoMix") was used as positive control, and nuclease-free water and CTAB buffer from the DNA extraction were used as negative controls. To quantify bacterial loads in plant samples (dataset 2) we performed a two-step host-associated microbe PCR (hamPCR) with simultaneous amplification of a plant single copy gene (GI = *GIGANTEA* gene, referred to as "GI gene") along with bacterial 16S rRNA genes according to[32] (primer sequences see Supplementary Tab. 4). With this protocol, bacterial 16S data can be normalized the GI reads to provide an estimate of bacterial loads in leaves. In all cases, libraries were prepared using a 2-step PCR protocol. In a first 5-cycle PCR, samples were amplified using 341 F/799 R "universal" 16S rRNA primers modified with an overhang sequence. Plant samples also contained blocking oligos to reduce plastid 16S amplification[74] and for dataset 2, GI primers, as mentioned above. The PCR product was enzymatically cleaned to remove remaining primers and used as template in a second, 35-cycle, PCR to add sample index barcodes and sequencing adapters using primers that bound to the overhang region (primer sequences see Supplementary Table 4). PCR products were cleaned up with magnetic beads and libraries were quantified using PicoGreen (1:200 diluted stock, Quant-iT™ PicoGreen™, ThermoFisher) in a qPCR machine (qTower³, JenaAnalytik, Jena, Germany) or by fluorescence on a gel using ImageJ. Samples were pooled according to their normalized fluorescence relative to the highest fluorescent sample. Pools from the hamPCR protocol with GI were further processed to increase the fraction of 16S relative to GI, as recommended in the original protocol. Libraries were sequenced on an Illumina MiSeq instrument for either 600 cycles (dataset 2, 3) or 300 cycles (dataset 1, 4). The precise procedures, including master mix recipes, thermocycling programs and primer sequences can be found in the Supplementary methods.

**Data analysis of amplicon sequencing.** For all four datasets we first split the amplicon sequencing data on indices and trimmed the adapter sequences from distal read ends using Cutadapt 3.5[75]. We then clustered amplicon sequencing data (forward reads only as they were much higher quality) into amplicon sequencing variants (ASVs) using

dada2[76]. We then removed chimeric sequences and retrieved a sequence table from the merged data. We assigned taxonomy to the final set of ASVs using the Silva 16S rRNA (v 138.1) database[77]. The database was supplemented by adding the *A. thaliana* GI gene sequence. All positive and negative controls for the datasets were checked. The distribution of taxa in the positive controls were as expected and the negative controls in all cases had <50 reads. In dataset 2 (leaf bacteriomes of lab-grown plants), several Zymomix ASVs (from the positive control) were observed in the negative control and other samples. Since this likely represented low-level background contamination detectable in samples with very low bacterial loads, Zymomix ASVs were removed prior to downstream analysis. We performed downstream analysis in R with Phyloseq[78] and VEGAN[79] for all data sets. If applicable, host-derived reads were removed by filtering any ASVs in the order "Chloroplast" and family "Mitochondria" from the 16S ASV tables. For dataset 2, plant GI reads were used to quantify the relative bacterial loads on the plant leaves. To do so, bacterial reads were normalized to the GI reads in each sample. This resulted in small fractions that were not usable with some downstream software, so it was scaled up by multiplying by a factor so that the smallest number in the abundance table is 1. In dataset 3, chloroplast reads were used to identify and remove Brassicales plants from non-*Arabidopsis* samples. Further analyses for specific datasets and for calculating richness, evenness, beta diversity, and differential abundance analysis are described in detail in the Supplementary methods.

### Statistical Analysis

All statistical analyszes which we performed are mentioned in the respective methods section and in the figure captions in the main text.

If not specified otherwise, all boxplots show the distribution of the data: The box shows the interquartile range (IQR) from the first to the third quartile, the line in the middle represents the median. The whiskers extend from the first/third quartile to 1.5× IQR. Outlier values above or below the whiskers' range are represented as dark dots.

### Reporting summary

Further information on research design is available in the Nature Portfolio Reporting Summary linked to this article.

## Data availability

The amplicon sequencing data generated in this study have been deposited in the NCBI-SRA database under accession codes: PRJNA1032255. PRJNA815825. PRJNA1124263. The bacterial genome sequencing data generated in this study have been deposited in the NCBI-SRA database under accession codes: PRJNA1124271. PRJNA1152919. All processed data with code to generate the main and Supplementary Figs. are available at Figshare in our folder: [https://figshare.com/projects/Glucosinolate_structural_diversity_shapes_recruitment_of_a_metabolic_network_of_leaf-associated_bacteria/180211]. Individual items are saved at figshare with their respective DOI numbers: [https://doi.org/10.6084/m9.figshare.26085514] Data & Script for Fig. 7C. [https://doi.org/10.6084/m9.figshare.26085520] Data & Script for Fig. 7G. [https://doi.org/10.6084/m9.figshare.26086114] Data & Script for Fig. 6C, D. [https://doi.org/10.6084/m9.figshare.24297421] Data & Scripts for Fig. 4 and Fig. S8, S9. [https://doi.org/10.6084/m9.figshare.26086120] Data & Script for Figs. 7E, F. [https://doi.org/10.6084/m9.figshare.26086123] Data & Script for Figs. 6A, 7A and S12. [https://doi.org/10.6084/m9.figshare.24242887] Data & Scripts for Fig. 7D and Fig. S13A. [https://doi.org/10.6084/m9.figshare.26022871] Data & Script for Fig. S14. [https://doi.org/10.6084/m9.figshare.26022958] Data for Fig. S10. [https://doi.org/10.6084/m9.figshare.24242881] Data & Scripts for Fig. 5 and S11. [https://doi.org/10.6084/m9.figshare.24297424] Data & Scripts for Fig. 3 and Fig. S5, S6, S7. [https://doi.org/10.6084/m9.figshare.26021752] Data S3 - Homology search in R3. [https://doi.org/10.6084/m9.figshare.24297385] Data

for Fig. 2D. [https://doi.org/10.6084/m9.figshare.24297409] Data & Scripts for Fig. 6B. [https://doi.org/10.6084/m9.figshare.24242812] Data for Fig. 1B. [https://doi.org/10.6084/m9.figshare.24242854] Data & Scripts for Fig. 2A and Fig. S2. [https://doi.org/10.6084/m9.figshare.24297379] Data & Scripts for Fig. 2B, C. [https://doi.org/10.6084/m9.figshare.24297427] Data for Fig. S1. [https://doi.org/10.6084/m9.figshare.24297502] Data for Fig. S3. [https://doi.org/10.6084/m9.figshare.24523660] Data for Tab S3.

## Code availability

R codes to generate the main figures are available at Figshare (see above) [https://figshare.com/projects/Beyond_defense_Glucosinolate_structural_diversity_shapes_recruitment_of_a_metabolic_network_of_leaf-associated_bacteria/180211].

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

## Acknowledgements

We acknowledge René Maskos, Stefan Riedel, and Kirsten Küsel (Aquatic Geomicrobiology, Friedrich-Schiller-University Jena) for sequencing our libraries on their MiSeq instrument. Additionally, we acknowledge the help of Beate Rothe in the Biochemistry Department at the Max-Planck-

Institute for Chemical Ecology with glucosinolate extractions. Funding was provided from the following sources: Carl Zeiss Foundation via Jena School for Microbial Communication (KU, TM, MTA). Deutsche Forschungsgemeinschaft (DFG, German Research Foundation) under Germany's Excellence Strategy - EXC 2051 - Projektnummer 390713860 (TM, MTA). Deutsche Forschungsgemeinschaft (DFG, German Research Foundation), Projektnummer 458884166 (MTA). International Max Planck Research School "Chemical Communication in Ecological Systems" (SAKR). Deutsche Forschungsgemeinschaft (DFG, German Research Foundation), Projektnummer 460684957 (UW, AH). Max Planck Society (JG, MR). JS received no particular funding for this work.

## Author contributions

Conceptualization: M.T.A., K.U. Methodology: K.U., T.M., S.A.K.R., M.R., A.H., J.S., M.T.A. Investigation: K.U., T.M., S.A.K.R., M.R., A.H., J.S. Visualization: K.U., S.A.K.R. Supervision: M.T.A., J.G., U.W. Writing - original draft: M.T.A., K.U. Writing - review & editing: all authors.

## Funding

## Competing interests

The authors declare no competing interests.
