## [Peer Review File · Nature Communications]

REVIEWER COMMENTS

Reviewer #1 (Remarks to the Author):

In this current manuscript by Unger et al., the authors described an unprecedented function of aliphatic glucosinolates (GLS) in a sidechain-dependent manner. Using leaf extracts from local natural *Arabidopsis* populations with different chemotypes, the authors found that allyl GLS positively affects the colonization of certain bacterial lineages. They isolated one bacterial strain that can utilize allyl GLS as the sole carbon source and that this catabolic process is a part of metabolic complementation across different bacterial strains *in vitro*. The manuscript is written in a clear, concise manner and provides an intriguing piece of information toward the mechanistic understanding of how glucosinolates regulate plant-microbiota interactions and, ultimately, how presumably antibiotic secondary metabolites could possibly control such interactions in multiple different ways. I would very much look forward to their future work, where they can knock out myrosinases from bacterial strains they have obtained and test their *in planta* role in leaf colonization.

The chemotypic comparison of Col-0 and NG2 revealed that NG2 predominantly accumulates allyl GLS and that its primary degradation product is CETP, an epithionitrile. On the other hand, the study mainly focused only on allyl GLS. I assume it is based on the assumption that plant-colonizing bacteria do not trigger cell collapse and the intact GLS play an important role, but I wonder to which extent this holds true. In fact, none of the enriched bacterial taxa in the order Burkholderiales was found to be metabolizing allyl GLS. This inconsistency may be due to different activities that CETP and allyl GLS might possibly have. CEPT feeding assay would be ideal, although I understand this is technically challenging. It would also be an interesting experiment to knock out ESP in NG2 to shift metabolic pathways from epithionitriles to isothiocyanates and see whether leaf microbiome profiles are altered or not. If it is indeed allyl GLS that plays the key role, there should not be any critical community shift by the loss of ESP function.

Related suggestion (not a request): Have the authors looked into the ESP sequence in the NG2 accession? As it is widely known that the ESP sequences and/or their expression patterns have been diverged among different *Arabidopsis* accession, it might be interesting to look into it to explain the molecular basis of CEPT production in NG2 leaves.

The authors also claimed that allyl GLS on the leaf surface is important for the leaf bacterial community assembly. However, there is no experimental evidence that intact allyl GLS is indeed deposited on the leaf surface. They also do not have data on the leaf surface microbial community structures, and it remains unclear where allyl GLS (or its degradation products) might actually work. The authors could have compared the leaf surface microbiome compositions between Col-0 and

NG2, as well as NG2 and NG2myb28, using the method they applied in Figure 5. One could also use DMSO to obtain the leaf wash and analyze its chemical composition, to directly support the notion that ally GLS is deposited on the leaf surface, although I am not entirely sure if that is technically possible.

Nevertheless, despite these uncertainties, it appears clear that ally GLS indeed has an impact on microbial community assembly. Leaf wash culture experiment is sound in this context, but the microbial profile of the starting inoculum is missing. Without knowing the initial community structure, one cannot claim the actual effect of the carbon sources added to the cultures. It is also disappointing that they performed experiments with only one starting inoculum, and it remains unclear how this result is reproducible and robust in different starting community structure.

Catabolism of 2OH3But-GLS and ally GLS by R3 and the synergistic growth between R3, Ps9, A, and J4 appear interesting. However, they only observed OD600 to measure "community growth" and did not test which of the strains in the mixture grew better in the co-culture setup. This could have been easily tested by 16S rRNA sequencing or simply measuring colony ratio after spreading on a growth plate. One could also search for an antibiotics that restrict the growth of either strain to more precisely quantify CFUs for each strain. This is of critical importance, as "better community growth" does not explain their enrichment in the leaf community.

In the same experiment, it would have been very interesting and exciting if the authors could measure the abundance of ally ITC, ally GLS, or other related metabolites after co-culturing. This is particularly helpful to address whether synergistic growth effect is due to metabolic complementation between different strains or rather mediated by yet another mechanism.

Growth rate quantification in figure 6E appears puzzling. They show that the growth rate of R3_A is way higher than all other conditions, but the growth curve in Figure 6D does not support this apparently. I do see a slight difference in how OD600 increases over time, but, for example, the slope of R3_A, R3_Ps, and R3_Ps9_J4 do not seem to be largely different, at least between 30 and 50 hours. The authors might wish to reconsider how they calculate the growth rate from this growth curve.

Lastly, they tend to generalize their claim at the taxon level, although their data is at the level of ASVs or strains. For example, in Figure 6, they mentioned that "Burkholderiales tend to positively influence community growth", while they only had two strains from Burkholderiales, belonging to two different families, which is not broad enough to infer a characteristic of the entire order. Interpretation of Figure 3 is also puzzling. They identified 13 ASVs to differentially colonize leaves of NG2 in a MYB28-dependent manner, and seven of them belonged to the order Burkholderiales; however, it was not entirely clear how the other Burkholderiales ASVs behave. If there are a way

greater number of Burkholderiales ASVs that did not show any significant difference, they should not generalize their findings to the entire order. Or, is it actually an aggregated abundance of each genus that is shown in Figure 3C? If that is the case, then the authors should analyze the dispersion of the abundance of ASVs within each taxon, in order to discriminate whether the enrichment is triggered by a few extreme ASVs or by the majority of respective ASVs. In silico depletion of Burkholderiales and re-computation of beta-diversity might also be helpful.

Some minor comments concerning the discussion:

The model in Figure 7 suggests that goitrin makes a negative feedback loop, explaining the slower growth of R3 in the presence of 2OH3But-GLS, but it was not clear whether goitrin has already been shown to be antimicrobial elsewhere.

The authors suggested that another bacterial population replaces the functional role of Yersiniaceae in planta, explaining why it was not enriched in the phyllosphere community they analyzed. However, the fact that they only isolated a single bacterial strain to be able to catabolize allyl GLS makes their model less likely. It needs to be better discussed.

Reviewer #1 (Remarks on code availability):

Some scripts are "not publicly available". However, in general, the codes are written clearly and concisely with appropriate explanations.

Reviewer #2 (Remarks to the Author):

The manuscript from Unger et al. reports a novel function of aliphatic glucosinolates (GLSs) in the interactions of the model plant *Arabidopsis thaliana* with endophytic bacteria. So far, aliphatic GLSs were mainly considered as precursors of biologically active isothiocyanates (ITCs), which can deter insects and restrict growth of microbial pathogens. To investigate the underestimated role of aliphatic GLSs in shaping leaf microbiome the authors applied metagenomics and targeted

metabolites to analyze wild populations of *A. thaliana* naturally occurring in the surrounding region. They also performed a series of in vitro assays to test growth of isolated endophytic bacteria on media containing plant homogenates or pure ITCs/GLSs. Presented results indicated that allyl-GLS, abundant in some of the analyzed *A. thaliana* wild accessions, but not in the model Col-0 ecotype, can serve as carbon source supporting growth of unique Yersiniaceae strain (R3), which in turn affects growth of other taxa that shape fitness of leaf bacterial community. These findings are interesting, and novel and the supporting evidence is firm and convincing.

Despite solid evidence for the function of allyl-GLS as carbon source, the proposed biochemical pathway supporting this phenomenon remains partially obscure. The authors assume that R3 strain utilizes own myrosinase to release glucose (as carbon source) from GLS molecule. The other product, potentially antimicrobial ITC, is supposed to be detoxified by bacterial Sax ITC-hydrolase to an amine (detected during analysis of culture media containing respective GLS; Fig. 6B). Theoretically, the same reactions should enable efficient usage of other GLSs as carbon source, but this is not the case. The authors show that ITC released upon 2OH3But-GLS hydrolysis rearranges to gotrin (not an ITC), which can be toxic, but is not a substrate of ITC-hydrolase. They also speculate that R3 cannot benefit from 4MSOB-GLS as a carbon source because toxic 4MSOB-ITC is not accepted by the putative Sax ITC-hydrolase due to its substrate specificity. However, it remains obscure if R3 indeed possesses an enzyme capable to convert allyl-ITC to an amine. The authors show presence of respective amine in the media, but there is no evidence that it is really formed from the corresponding ITC. Concerning the complexity of glucosinolate degradation, it remains possible that other enzymes/intermediates are involved in formation of the amine. The hypothetical ITC-hydrolase activity, together with its specificity, could be easily tested by cultivating R3 strain on media containing 4MSOB-ITC and allyl-ITC (as in Fig. 2BC and Fig. S3).

Discussing results of growth assays in plant hydrolysates the authors conclude “These opposing effects suggest that aliphatic GLS breakdown products from different *A. thaliana* genotypes act very differently towards bacteria” (lines 50-52). It should be considered that NG2 and Col-0 may differ not only in accumulation of aliphatic GLS, but also of other metabolites. Some of such other metabolites can also contribute to the discussed “opposing effects”. In the same paragraph the authors note “Interestingly, NG2 leaf extract medium was less inhibitory, and several strains grew significantly more than in Col-0 myb28/myb29 extract”. This indicates that apart from hydrolysis products of aliphatic GLSs Col-0 extract contains additional antimicrobial metabolites that are absent or less abundant in NG2. Impact of such metabolites should be considered when the differences between Col-0 and NG2 microbiomes are discussed.

To investigate the role of aliphatic GLSs in the assembly of leaf microbiota the authors applied CRISPR-Cas9 to generate myb28 mutant in the NG2 ecotype. Strikingly, unlike myb28 mutant in Col-0, this mutant was completely deficient in aliphatic GLS production. In Col-0 such strong metabolic phenotype is observed only if both MYB28 and MYB29 are simultaneously mutated (Fig. 1B). This suggests that (i) in NG2 ecotype MYB29 does not contribute to the biosynthesis of aliphatic

GLSs, or (ii) due to the homology between MYB28 and MYB29 Cas9 damaged also MYB29 gene in the obtained NG2 myb28 mutant. Did the authors check MYB29 gene in NG2 and in NG2 myb28 plants? It does not affect the authors conclusions made based on the results obtained with this line, but would be of interest to clarify if this is a single myb28 or a double myb28 myb29 mutant.

l. 74-75 “The chemical diversity of GLSs is determined by their side chains which result from their biosynthesis from different amino acids”. This is only one of the factors contributing to GLS diversity. This diversity also results from precursor amino acid modification (e.g. Met chain elongation) and glucosinolate modification. It would be of interest to briefly explain this, especially that the observed differences in aliphatic GLS profiles between tested ecotypes (Fig. 1B) result from such modifications. Currently, the non-expert reader can get an impression that the discussed aliphatic GLSs are derived from different amino acids, which is not true. Additionally, it could be of support if the authors include structures of the main discussed GLSs and their degradation products in a supplementary figure. Some of these structures are shown in Fig. 6B, but it would be of interest to show structures of some other discussed compounds (e.g 4MSOB-GLS or CETP). Moreover, the reaction schemes presented in Fig. 6B are somehow misleading. ITCs are not direct GLS hydrolysis products. A proper scheme of GSL hydrolysis could be also included in a supplementary figure.

l. 85 Myrosinases hydrolyze glucosinolates, not glucose. Glucose is one of the products of myrosianse-catalyzed GLS hydrolysis.

The font size in graph legends is relatively small and hard to read. I realize that in many cases it is rather difficult to increase it. However, it could be done in some figures (e.g. Fig. 4A).

Reviewer #3 (Remarks to the Author):

“Extract media” of Arabidopsis plants of the genotype NG2 supported more microbes than those of Col-0 plants. This is proposed to be due to glucosinolates, particularly allyl-GLS, in NG2 that positively influence microbial abundance. On leaves and also in media with allyl-GLS as a carbon source, Burkholderiales were enriched. Only a Yersiniaceae strain (R3) could grow on pure allyl-GLS, and this cross-fed other bacteria. The paper combines microbial analysis of field plants with increasingly defined media and synthetic communities to uncover to the level of individual bacterial strains how glucosinolates, major defensive molecules of Brassicaceae, affect bacterial growth, and how plant genotype differences affect microbial communities through GLS profiles.

The paper could use better graphical organization, and perhaps would benefit from one additional experiment (first comment below).

Line numbers on the PDF are cut so it's not possible unfortunately to always refer to the correct line.

>> Given the surprising result that NGmyb28 plants had lower bacterial loads than NG2 plants, it would be worth also testing NGmyb28 extract media vs. NG2 extract media to see if this is the case here as well, and if similar microbes are enriched. It does not seem this experiment has been done. In Figure 2A, extract media for NG2, myb28/29, myb28, and Col-0 are shown; it is not clear what the parent genotype of the mutants is, but it seems to be Col-0 from the color scheme. This should also be made more clear.

>>The other ruderal plants are never defined. There is also not a picture of them. In the supplementary material they are also incorrectly called "random". Can the authors provide any more information on what these plants were, or even just some of the plants that this included? Many times other small Brassicaceae can be found among *A. thaliana*. I suppose the authors purposely excluded these, so as to avoid glucosinolates?

>>It's unclear how the leaf passages are performed. A diagram or more step by step guide would help. Was the media shaken / agitated? The text says intervals were adjusted for growth rate differences. "It took seven days for"... it took 7 days for what? Sufficient cloudiness? How cloudy was sufficiently cloudy? In one part of the text it says "passages on 4MSOB-GLS medium", and refers to cell density, which doesn't make sense in the context of a plate. The "on" suggests a plate. The text should be clear always in referring to liquid media or solid media. "plates" and "tubes" would also be acceptable ways of making the distinction between media types.

>>Fig 3 C... these are differentially abundant taxa according to the legend, but the p values of Wilcoxon tests for some of the comparisons are in the range that would not be considered significant (eg. 0.18). If the initial call of significance is based on a different test, why was the Wilcoxon test also done and which test is better?

>>Fig 3C... leaves of *Arabidopsis* often have *Methylobacteria*, *Pseudomonas*, *Rhizobia*, *Sphingomonas*, and others... but the microbes listed here are generally of lesser abundance. Yet, these apparently are the microbes driving the lower microbial load in the NGmyb28 mutant? Perhaps it's worth comparing the overall microbial community visually for NG2 and for NGmyb28 (eg the stacked column chart like Fig 1B, but showing bacterial families instead of GLS species).

>>fig 4B.. it's odd and difficult to track, having the genera indicated by the point color. Perhaps the genera could be indicated by the color of the box at the top of each chart containing the genus name. Also, some of the names are cut by the edges of the box...

>>Fig 5 is fairly difficult to see in terms of font sizes. Why was there only a single technical replicate of 4MSOB-GLS? Why does the experiment with glucose followed by allyl-GLS say n=3 but shows only 2 technical replicates?

>>The taxa recovered by amplicon sequencing are said to be similar to the taxa growing with 4MSOB-GLS and allyl-GLS media, and Tab S3 and Fig. 5A are mentioned. It would be helpful to see the information about which microbes could be cultured from each media also somehow in the same figure 5A.

>>”Burkholderiales tend to positively influence community growth”... what does this mean? What data supports this? I don't recall an experiment in which the addition of Burkholderiales led to the growth of multiple other bacteria.. or am I missing something? I see the point that they are around as long as R3 or another allyl-GLS metabolizer is there. Is this based on literature reports of them being “friendly”? If so, that has not yet been shown for the isolates tested in the study, so probably should not be stated so boldly.

>>This sentence is unclear “Assuming this enrichment is indeed due to GLS metabolism, it is possible that bacteria that can access GLSs do not need to be present in high abundances so that we did not detect Yersiniaceae”.

Reviewer #1 (Remarks to the Author):

In this current manuscript by Unger et al., the authors described an unprecedented function of aliphatic glucosinolates (GLS) in a sidechain-dependent manner. Using leaf extracts from local natural *Arabidopsis* populations with different chemotypes, the authors found that allyl GLS positively affects the colonization of certain bacterial lineages. They isolated one bacterial strain that can utilize allyl GLS as the sole carbon source and that this catabolic process is a part of metabolic complementation across different bacterial strains *in vitro*. The manuscript is written in a clear, concise manner and provides an intriguing piece of information toward the mechanistic understanding of how glucosinolates regulate plant-microbiota interactions and, ultimately, how presumably antibiotic secondary metabolites could possibly control such interactions in multiple different ways. I would very much look forward to their future work, where they can knock out myrosinases from bacterial strains they have obtained and test their *in planta* role in leaf colonization.

We thank the reviewer for their positive encouragement

The chemotypic comparison of Col-0 and NG2 revealed that NG2 predominantly accumulates allyl GLS and that its primary degradation product is CETP, an epithionitrile. On the other hand, the study mainly focused only on allyl GLS. I assume it is based on the assumption that plant-colonizing bacteria do not trigger cell collapse and the intact GLS play an important role, but I wonder to which extent this holds true. In fact, none of the enriched bacterial taxa in the order Burkholderiales was found to be metabolizing allyl GLS. This inconsistency may be due to different activities that CETP and allyl GLS might possibly have. CEPT feeding assay would be ideal, although I understand this is technically challenging.

The reviewer is correct that we assumed that CETP would not be present in large amounts in healthy leaves because we did not expect leaf damage due to these bacteria, and leaf damage in the presence of the epithiospecifier protein ESP, would be required to form CETP when allyl-GLS is hydrolyzed. However, since we currently don't have this chemical resolution, we addressed this by growing the representative bacterial strains from the enrichment experiments and other bacteria in the presence of CETP. The reviewer is right that it is not easily available, so we instead used a crushed leaf medium. Since leaf homogenates of the WT NG2 contains CETP (original figure 2D), we expected if the bacteria can benefit from CETP or other trace products, they should grow better there than in the NG2 *myb28* mutant leaf medium. Only one bacterium, a *Xanthomonas*, grew slightly better in NG2 WT medium while four, including an isolate recovered in the enrichment experiments, showed decreased growth, suggesting that CETP and/or other GLS breakdown in NG2 can even have toxic effects. This provides additional justification of focusing on the role of the GLS in enrichment.

We have included this data as Fig S14 and it is described in the text at line 297. The methods were also updated accordingly.

It would also be an interesting experiment to knock out ESP in NG2 to shift metabolic pathways from epithionitriles to isothiocyanates and see whether leaf microbiome profiles are altered or not. If it is indeed allyl GLS that plays the key role, there should not be any critical community shift by the loss of ESP function.

We agree that this would be a nice extra experiment. However, generation of mutants in the NG2 line is very time-intensive and given the new results (Fig S14) showing that GLS breakdown products do not benefit growth of the bacteria, we prefer to focus on GLS in the current manuscript.

Related suggestion (not a request): Have the authors looked into the ESP sequence in the NG2

accession? As it is widely known that the ESP sequences and/or their expression patterns have been diverged among different Arabidopsis accession, it might be interesting to look into it to explain the molecular basis of CEPT production in NG2 leaves.

We agree that this would be interesting to look into. We do have *A. thaliana* NG2 WT genomic data, but this would likely require comparing expression data to Col-0 which is beyond the scope of this manuscript. Thus, since this is not a request we prefer to leave this for future analysis.

The authors also claimed that allyl GLS on the leaf surface is important for the leaf bacterial community assembly. However, there is no experimental evidence that intact allyl GLS is indeed deposited on the leaf surface. They also do not have data on the leaf surface microbial community structures, and it remains unclear where allyl GLS (or its degradation products) might actually work. The authors could have compared the leaf surface microbiome compositions between Col-0 and NG2, as well as NG2 and NG2myb28, using the method they applied in Figure 5. One could also use DMSO to obtain the leaf wash and analyze its chemical composition, to directly support the notion that allyl GLS is deposited on the leaf surface, although I am not entirely sure if that is technically possible.

Regarding the leaf surface bacteriomes, we feel that this point arises from a lack of clarity in the manuscript. We did measure bacterial communities in both whole leaves (surface bacteria + endophytes) and in surface-sterilized leaves (endophytes). The main difference between these is the removal of surface bacteria. The bacterial loads in whole leaves of NG2 and NG2myb28 were much higher than endophytes (Fig 3A), suggesting that the majority of colonizers were surface bacteria. Additionally, we only saw effects between WT and aliphatic-GLS-free mutants in whole leaves of NG2 (Fig 3B – whole leaves, Fig S5 – endophytes and whole leaves, Data S2 – significance table). In Col-0, we did not observe an effect of the lack of GLS, which indicates that this effect of the removal of GLSs is unique to NG2. Together, this strongly indicates that mainly surface bacteria were affected by the presence of GLSs in NG2. We have added a sentence at line 180 to try to make this point clearer.

Regarding GLS presence on the surface, GLS have previously been confirmed to be present on the leaf surface using MS imaging, which we cited in our discussion (Shroff et al. 2015, l 347). Additionally, their biosynthesis has been confirmed in trichomes, which are known to secrete secondary metabolites (Frerigmann et al. 2012, [10.3389/fpls.2012.00242](https://doi.org/10.3389/fpls.2012.00242), l 348). However, to ensure that GLS were present on leaves in our hands, we now performed an additional experiment by dipping leaves of Col-0, NG2, and their GLS-free mutants in methanol and evaluating these surface extracts for GLSs, similar to methods used in previous studies (de Vos et al. 2008, [10.1104/pp.107.112185](https://doi.org/10.1104/pp.107.112185), Suppl. Methods). In all wild-type plants, we were able to detect the expected major aliphatic glucosinolates (Col-0 = 4MSOB-GLS, NG2 = allyl-GLS). Thus, we conclude that it is plausible that leaf surface bacteria are exposed to leaf surface GLSs. We have described this new data as Fig S10 and included it at line 225. We updated the supplementary methods accordingly and related Tab. S6.

Nevertheless, despite these uncertainties, it appears clear that allyl GLS indeed has an impact on microbial community assembly. Leaf wash culture experiment is sound in this context, but the microbial profile of the starting inoculum is missing. Without knowing the initial community structure, one cannot claim the actual effect of the carbon sources added to the cultures.

We tried, but unfortunately we used our standard DNA extraction protocol to extract the inoculum and were not successful in recovering enough DNA, which reflects the low amounts of bacteria that would be washed off of leaves. We would have probably needed to use a method designed to recover small amounts of DNA. However, we have tried to address this another way, since we have community information from each passage of the enrichment experiments. Here, we can clearly see

that in the case where bacteria were first grown on glucose then passaged to allyl-GLS, the community quickly took on a structure similar to the allyl-GLS-only enrichments, even though after glucose the community was very different. The communities that continue on glucose are very different. So this shows enrichment specifically due to allyl-GLS. We have now added this data to the manuscript in detail in an additional figure (Fig S11). It is also possible to see in Fig 5B that after one passage on glucose the *Oxalobacteraceae* are still very low abundance (or below detection) but that they increase significantly when this is switched to allyl-GLS, but not when it continues on glucose. Thus, the structure observed after three passages is very specific to allyl-GLS as the sole carbon source. We have adjusted the text at line 244 and line 250 to address this in the text more clearly.

It is also disappointing that they performed experiments with only one starting inoculum, and it remains unclear how this result is reproducible and robust in different starting community structure.

It is true that the two specific allyl-GLS community structures that we enriched probably represent only two possible "outcomes". However, we feel that this one "case" still accomplishes what is needed at this point in the manuscript - a tool for us to dissect some mechanisms behind what we observed in the plants. These are: 1) Enrichment can be explained by utilization of GLS as a C-source, 2) Testing whether enrichment reflects chemotype specificity observed in plants and identifying a likely mechanism for specific enrichment on allyl-GLS but not 4MSOB-GLS and 3) Given few taxa apparently use allyl-GLS, showing that exchange of metabolites from a key organism to others can explain enrichment of diverse taxa. Thus, while we agree in principle that adding more inocula would add insight by for example showing if other community configurations assemble on allyl-GLS, it would be a large undertaking in terms of time and expense, and we instead invested significant resources for this revision into addressing those three key questions more robustly and in more detail (see below), and did not add more replicates at this time.

Regarding reproducibility, we ask the reviewer to instead refer to the consistency in the other parts of the manuscript: members of *Oxalobacteriaceae* and *Comamonadaceae*, as well as a few other taxa were consistently observed to be enriched in association with GLS in NG2. That includes in the lab experiments where plants recruited bacteria from a natural soil, in two years of observation of wild NG2 plants, where they were enriched in *Arabidopsis* compared to other non-Brassicaceae plants, and in this enrichment experiment where *Janthinobacterium* (*Oxalobacteriaceae*) was always enriched in multiple replicate enrichments, performed in multiple ways, from randomly collected wild NG2 leaves. We feel that this is strong evidence that these and other bacteria were enriched in association with allyl-GLS, and we are excited that our other revisions provide stronger insights into how this enrichment works. We did work on the discussion starting at L395 to try to more clearly reiterate the consistency of the recruitment of these taxa and what that might mean. Please note we have also made changes throughout the text to try to improve focus on the key questions outlined above, for example by now dividing Fig 6 (addressing question 2) and Fig 7 (addressing question 3) .

Catabolism of 2OH3But-GLS and allyl GLS by R3 and the synergistic growth between R3, Ps9, A, and J4 appear interesting. However, they only observed OD600 to measure "community growth" and did not test which of the strains in the mixture grew better in the co-culture setup. This could have been easily tested by 16S rRNA sequencing or simply measuring colony ratio after spreading on a growth plate. One could also search for an antibiotic that restricts the growth of either strain to more precisely quantify CFUs for each strain. This is of critical importance, as "better community growth" does not explain their enrichment in the leaf community.

In the same experiment, it would have been very interesting and exciting if the authors could measure the abundance of allyl ITC, allyl GLS, or other related metabolites after co-culturing. This is

particularly helpful to address whether synergistic growth effect is due to metabolic complementation between different strains or rather mediated by yet another mechanism.

In response to both of these comment and the comment below from another reviewer arguing that the improvements in growth rate were limited, we scrapped the original experiment and started over. In the new setup, we used the experiment only to test whether and how multiple taxa can become enriched with *Rahnella* R3 on allyl-GLS. Specifically, we tested combinations of *Rahnella*, *Pseudomonas*, *Stenotrophomonas* and *Janthinobacterium* in culture on allyl-GLS as C-source and confirm that *Rahnella* is required for growth. In a longer-term three-passage experiment with combinations of *Rahnella*, *Pseudomonas*, and *Janthinobacterium*, we used amplicon sequencing to check their persistence after three passages and untargeted and targeted metabolomics to determine how they affected the metabolome compared to *Rahnella* alone. Figure 7 now focuses on these questions, showing that a) *Rahnella* is required for community growth, b) *Pseudomonas* grows quickly on the spent medium of *Rahnella*, c) *Pseudomonas* and *Janthinobacterium* are maintained through the passages in abundances similar to those in the enrichments from wild leaves, and d) Metabolomics suggests that *Pseudomonas* depletes a number of metabolites associated with *Rahnella*, and *Janthinobacterium* addition tends to further decrease these metabolites. This is strong evidence that these bacteria can persist by cross-feeding from *Rahnella*. As a bonus, we also observed a decrease in the residual allyl-ITC in more diverse cultures, at levels that are likely to be relevant for toxicity and physiological effects. Thus, multiple bacteria appear to both feed from *Rahnella* and potentially benefit the community as a whole. Please refer to Figure 7, Supp Figs 12 and 13 and the completely revise accompanying text with Figure 7. The new experiments have been described in detail in the methods.

Growth rate quantification in figure 6E appears puzzling. They show that the growth rate of R3_A is way higher than all other conditions, but the growth curve in Figure 6D does not support this apparently. I do see a slight difference in how OD600 increases over time, but, for example, the slope of R3_A, R3_Ps, and R3_Ps9_J4 do not seem to be largely different, at least between 30 and 50 hours. The authors might wish to reconsider how they calculate the growth rate from this growth curve.

We agree that the differences were hard to see in Figure 6D, which in part was because we plotted the whole time of the experiment and so it is hard to see the differences in the early active growth time. However, it is true that the rate effect, while significant, was not a huge one for any of the added strains. Thus, we scrapped this experiment in favor of the deeper analysis described in the response above. This now better addresses the core idea that allyl-GLS recruitment involved a metabolic network of bacteria.

Lastly, they tend to generalize their claim at the taxon level, although their data is at the level of ASVs or strains. For example, in Figure 6, they mentioned that "Burkholderiales tend to positively influence community growth", while they only had two strains from Burkholderiales, belonging to two different families, which is not broad enough to infer a characteristic of the entire order.

We agree that we tended to be a bit too general about Burkholderiales which may cause confusion. In response to this and the next comment, we analyzed enrichment at higher levels and realized that indeed the entire Burkholderiales order is also enriched in the lab experiments NG2 vs NG2 *myb28*. However, Burkholderiales is a large order and we see enrichment of mainly a few families and genera. Thus, we went throughout the text and everywhere we generally talked about enrichment of Burkholderiales, we now more specifically refer to these taxa. The changes are marked in the text associated with the lab experiment, the wild population data, the in-vitro enrichment experiment and

in the discussion (for example L397). The family and order-level analyses are provided in Fig S6 (lab experiment) and in Fig S9 (wild plants).

Interpretation of Figure 3 is also puzzling. They identified 13 ASVs to differentially colonize leaves of NG2 in a MYB28-dependent manner, and seven of them belonged to the order Burkholderiales; however, it was not entirely clear how the other Burkholderiales ASVs behave. If there are a way greater number of Burkholderiales ASVs that did not show any significant different, they should not generalize their findings to the entire order. Or, is it actually an aggregated abundance of each genus that is shown in Figure 3C? If that is the case, then the authors should analyze the dispersion of the abundance of ASVs within each taxon, in order to discriminate whether the enrichment is triggered by a few extreme ASVs or by the majority of respective ASVs. In silico depletion of Burkholderiales and re-computation of beta-diversity might also be helpful.

Again, we agree that we unintentionally tended to over-generalizing the effect on Burkholderiales. In order to address this, we now analyzed the data at the family and order level in addition to the genus level (Fig S6). We find that the order Burkholderiales is highly enriched in NG2 relative to NG*myb28*, but this is due to only specific families and genera. We decided not to analyze at the ASV level, because in our experience ASVs are subject to artificial splitting due to sequencing errors. Thus, we typically feel that it is more conservative and justified to agglomerate data at the genus level. More fine-grained analysis will likely require future work that may include sequenced strains and/or metagenomics. We also checked our wild *A. thaliana* data and agglomerated on order and family level there (Fig. S9). There enrichment at higher levels was found only in the *Methylophilaceae* and *Comamonadaceae* families, underscoring that it is specific taxa that are enriched due to GLS and in the more diverse wild plant bacteriomes, enrichment of these specific taxa does not result in enrichment of the whole families. We include a description of these new results in line 187 and 209 and try now to stress everywhere it comes up that what was common among the three experiments were the enrichment of specific genera that belong to a few specific families within the Burkholderiales.

Some minor comments concerning the discussion:

The model in Figure 7 suggests that goitrin makes a negative feedback loop, explaining the slower growth of R3 in the presence of 2OH3But-GLS, but it was not clear whether goitrin has already been shown to be antimicrobial elsewhere.

In response to this and other reviewer comments, we checked whether goitrin and 4MSOB-ITC have a negative effect on R3. We found that the effect is limited, and R3 can even metabolize 4MSOB-ITC to 4MSOB-amine (see new Fig. 6), suggesting that our hypothesis was incorrect. It is likely therefore that the specificity arises at the step of the myrosinase hydrolysis. Thus, we have simplified the figure to remove complicated arrows and rather focus on our hypothesis for where the specificity arises. See new Fig 8. We have also adjusted the text and discussion accordingly.

The authors suggested that another bacterial population replaces the functional role of Yersiniaceae in planta, explaining why it was not enriched in the phyllosphere community they analyzed. However, the fact that they only isolated a single bacterial strain to be able to catabolize ally GLS makes their model less likely. It needs to be better discussed.

We agree that this was not clear. We rewrote the paragraph at line 372 to simplify it and clarify the message.

Reviewer #1 (Remarks on code availability):

Some scripts are "not publicly available". However, in general, the codes are written clearly and concisely with appropriate explanations.

We updated the R scripts and adjusted the data files to the revised version and everything should be publicly available now.

Reviewer #2 (Remarks to the Author):

The manuscript from Unger et al. reports a novel function of aliphatic glucosinolates (GLSs) in the interactions of the model plant *Arabidopsis thaliana* with endophytic bacteria. So far, aliphatic GLSs were mainly considered as precursors of biologically active isothiocyanates (ITCs), which can deter insects and restrict growth of microbial pathogens. To investigate the underestimated role of aliphatic GLSs in shaping leaf microbiome the authors applied metagenomics and targeted metabolomics to analyze wild populations of *A. thaliana* naturally occurring in the surrounding region. They also performed a series of in vitro assays to test growth of isolated endophytic bacteria on media containing plant homogenates or pure ITCs/GLSs. Presented results indicated that allyl-GLS, abundant in some of the analyzed *A. thaliana* wild accessions, but not in the model Col-0 ecotype, can serve as carbon source supporting growth of unique Yersiniaceae strain (R3), which in turn affects growth of other taxa that shape fitness of leaf bacterial community. These findings are interesting, and novel and the supporting evidence is firm and convincing.

We thank the reviewer for their positive encouragement.

Despite solid evidence for the function of allyl-GLS as carbon source, the proposed biochemical pathway supporting this phenomenon remains partially obscure. The authors assume that R3 strain utilizes own myrosinase to release glucose (as carbon source) from GLS molecule. The other product, potentially antimicrobial ITC, is supposed to be detoxified by bacterial Sax ITC-hydrolase to an amine (detected during analysis of culture media containing respective GLS; Fig. 6B). Theoretically, the same reactions should enable efficient usage of other GLSs as carbon source, but this is not the case. The authors show that ITC released upon 2OH3But-GLS hydrolysis rearranges to goitrin (not an ITC), which can be toxic, but is not a substrate of ITC-hydrolase. They also speculate that R3 cannot benefit from 4MSOB-GLS as a carbon source because toxic 4MSOB-ITC is not accepted by the putative Sax ITC-hydrolase due to its substrate specificity. However, it remains obscure if R3 indeed possesses an enzyme capable to convert allyl-ITC to an amine. The authors show presence of respective amine in the media, but there is no evidence that it is really formed from the corresponding ITC. Concerning the complexity of glucosinolate degradation, it remains possible that other enzymes/intermediates are involved in formation of the amine. The hypothetical ITC-hydrolase activity, together with its specificity, could be easily tested by cultivating R3 strain on media containing 4MSOB-ITC and allyl-ITC (as in Fig. 2BC and Fig. S3).

This was a good idea, and helped us a lot to better explain what we are seeing. Cultivation of R3 with 4MSOB-ITC, allyl-ITC and goitrin was done (new Fig. 6C) and we observed the formation of the corresponding amines, whereas goitrin was not degraded (new Fig. 6D). In addition, we sequenced the genome of R3 and searched for homologs of known and functionally tested SaxA proteins (described in Van den Bosch et al. 2018, DOI: [10.1128/AEM.00478-18](https://doi.org/10.1128/AEM.00478-18)), as well as for characterized bacterial myrosinases which were isolated from plant-associated natural sites (plants, soil). In the genome of R3 we found only two putative homologs of SaxA and 10 putative myrosinases. Two myrosinase homologs that are very similar to known ones also have secretion signals, likely important in the phyllosphere (see new Supplementary Data 3). We added this information (starting at L271), and the method in the supplementary and we split the old figure 6 into two new figures. The first one (new Fig. 6 = old Fig. 6A,6B) shows our results about the substrate specificity of R3 and includes the new data (6C-D). The second one (new Fig. 7) includes a new experiment digging into the community dynamics based on R3's metabolization of allyl-GLS. Together, it seems likely that a functional SaxA protein is expressed by R3, but we revised our hypothesis because the data did not support toxicity of GLS breakdown products like ITCs as the decisive factor for R3 growth with different GLSs. Rather we

hypothesize that the selectivity probably arises at the hydrolysis stage, possibly due to myrosinase specificity. We have also addressed this in the discussion (new Fig. 8 and accompanying text).

Discussing results of growth assays in plant hydrolysates the authors conclude “These opposing effects suggest that aliphatic GLS breakdown products from different *A. thaliana* genotypes act very differently towards bacteria” (lines 50-52). It should be considered that NG2 and Col-0 may differ not only in accumulation of aliphatic GLS, but also of other metabolites. Some of such other metabolites can also contribute to the discussed “opposing effects”.

We agree there are other differences than aliphatic GLSs between NG2 and Col-0 genotypes that might affect bacterial growth in leaf extract medium. We reworded this part in L 142. To get a clearer picture of the effects of aliphatic GLS breakdown products in both genotypes we repeated the bacterial growth assay in leaf extract medium with not just Col-0 and *myb28/29* extract medium but also leaf extract medium from NG2 and *NGmyb28* which was not available when we did the first leaf extract assay. We added this result as Fig. S14 and address this comment together with a comment from Reviewer #3 in line 297. We also added the methods accordingly. As expected, most tested isolates show significantly better growth in *myb28/29* medium compared to Col-0 medium with aliphatic GLS breakdown products. Especially commensal strains like *Plantibacter*, *Rhodococcus*, *Acidovorax*, *Janthinobacterium* or *Microbacterium* did not grow well in the WT medium. Fan et al. 2011 (DOI: [10.1126/science.1199707](https://doi.org/10.1126/science.1199707)) performed a bio-guided fractionation of Col-0 leaf extracts which were produced similarly to our leaf extract media and found 4MSOB-ITC to be the major inhibitory compound for the model bacteria *Escherichia coli*. Therefore, we hypothesize that our strains are affected in a similar manner. In NG2, however, we find only some strains to benefit from the loss of aliphatic GLSs in *NGmyb28* compared to NG2. Additionally, the bacterial growth differences between the WT and aliphatic GLS-free leaf medium are overall smaller than the differences in the Col-0 background, showing a reduced effect of aliphatic GLS breakdown products in NG2 background.

In the same paragraph the authors note “Interestingly, NG2 leaf extract medium was less inhibitory, and several strains grew significantly more than in Col-0 *myb28/myb29* extract”. This indicates that apart from hydrolysis products of aliphatic GLSs Col-0 extract contains additional antimicrobial metabolites that are absent or less abundant in NG2. Impact of such metabolites should be considered when the differences between Col-0 and NG2 microbiomes are discussed.

We agree, we did not consider other chemical differences between Col-0 and NG2 in this interpretation. We added text at L142 and L161 to more clearly explain that although there are probably other differences too, why we chose to focus our hypothesis in aliphatic GLS. Later in the text, we added another experiment where we directly compare NG2 to *NGmyb28* and Col-0 to *myb28/29* leaf extract medium to show the differences which correlate to the loss of aliphatic GLSs more specifically (Fig S14, see also previous comment).

To investigate the role of aliphatic GLSs in the assembly of leaf microbiota the authors applied CRISPR-Cas9 to generate *myb28* mutant in the NG2 ecotype. Strikingly, unlike *myb28* mutant in Col-0, this mutant was completely deficient in aliphatic GLS production. In Col-0 such strong metabolic phenotype is observed only if both MYB28 and MYB29 are simultaneously mutated (Fig. 1B). This suggest that (i) in NG2 ecotype MYB29 does not contribute to the biosynthesis of aliphatic GLSs, or (ii) due to the homology between MYB28 and MYB29 Cas9 damaged also MYB29 gene in the obtained NG2 *myb28* mutant. Did the authors check MYB29 gene in NG2 and in NG2 *myb28* plants? It does not affect the authors conclusions made based on the results obtained with this line, but would be of interest to clarify if this is a single *myb28* or a double *myb28 myb29* mutant.

We were also curious, so we sequenced the target site on the *myb29* gene and compared the sequence of our mutant NG*myb28* to the WT NG2. We observed no differences within this site: no truncation, no frame-shift mutation or exchange of individual bases. We still would need to check if this gene in NG2 genotype is in general different to *myb29* in Col-0 genotype, for example a natural KO mutation occurs, explaining why only the knock-out of *myb28* is sufficient to prevent formation of aliphatic GLSs in NG2 leaves. This scenario seems likely. Since this is not really relevant for the manuscript currently, we have not included it at this time.

I. 74-75 “The chemical diversity of GLSs is determined by their side chains which result from their biosynthesis from different amino acids”. This is only one of the factors contributing to GLS diversity. This diversity also results from precursor amino acid modification (e.g. Met chain elongation) and glucosinolate modification. It would be of interest to briefly explain this, especially that the observed differences in aliphatic GLS profiles between tested ecotypes (Fig. 1B) result from such modifications. Currently, the non-expert reader can get an impression that the discussed aliphatic GLSs are derived from different amino acids, which is not true.

We agree, we were not precise about the biosynthesis of GLS and now reworded this part (I 75) to make it more clear for all readers.

Additionally, it could be of support if the authors include structures of the main discussed GLSs and their degradation products in a supplementary figure. Some of these structures are shown in Fig. 6B, but it would be of interest to show structures of some other discussed compounds (e.g 4MSOB-GLS or CETP). Moreover, the reaction schemes presented in Fig. 6B are somehow misleading. ITCs are not direct GLS hydrolysis products. A proper scheme of GSL hydrolysis could be also included in a supplementary figure.

To clarify this part as proposed we visualized GLS hydrolysis in a schematic drawing in Fig. S1A. Additionally, we provide chemical structures of other relevant molecules (allyl-GLS, allyl-ITC, allyl-CN, CETP, 4MSOB-GLS, 4MSOB-ITC, 4MSOB-CN, 2OH3But-GLS, goitrin) in Fig. S1B. We link to these two additional figures in the main text (I 89 and in the legend of Fig. 2).

I. 85 Myrosinases hydrolyze glucosinolates, not glucose. Glucose is one of the products of myrosinase-catalyzed GLS hydrolysis.

We corrected the wrong wording (I 85).

The font size in graph legends is relatively small and hard to read. I realize that in many cases it is rather difficult to increase it. However, it could be done in some figures (e.g. Fig. 4A).

Thank you for this comment. We increased the font sizes in figures 4, 5 and the new figures 6, 7.

Reviewer #3 (Remarks to the Author):

“Extract media” of Arabidopsis plants of the genotype NG2 supported more microbes than those of Col-0 plants. This is proposed to be due to glucosinolates, particularly allyl-GLS, in NG2 that positively influence microbial abundance. On leaves and also in media with allyl-GLS as a carbon source, Burkholderiales were enriched. Only a Yersiniaceae strain (R3) could grow on pure allyl-GLS, and this cross-fed other bacteria. The paper combines microbial analysis of field plants with increasingly defined media and synthetic communities to uncover to the level of individual bacterial strains how glucosinolates, major defensive molecules of Brassicaceae, affect bacterial growth, and how plant genotype differences affect microbial communities through GLS profiles. The paper could use better graphical organization, and perhaps would benefit from one additional experiment (first comment below).

Thank you for this comment. We realize that our organization may have given some misconception, for example that we aimed to find out what made bacteria grow better in extract media. Actually, the extract media should mostly have GLS breakdown products, not intact GLS, and that was just a first indication of differences in secondary metabolites between the plant genotypes. Based on this, we had at that point actually expected an effect on the bacteriome in Col-0, and less effect in NG2. We have now made significant revisions (especially a major graphical revision of Figure 6, by dividing it into two sections with new experiments) to focus on the main messages: 1) Specific glucosinolates, surprisingly, are strongly correlated to the recruitment of specific taxa in leaves in both lab experiments and in the wild, 2) use of specific GLS as a carbon source can explain recruitment of the bacteria, 3) The specificity of recruitment for certain GLS structures (and therefore specific plant ecotypes) seems to arise at the myrosinase hydrolysis step and 4) Cross-feeding among bacteria can explain consistent recruitment of some taxa who cannot directly utilize glucosinolates. We hope that these changes make the main theme – recruitment to intact GLS in leaves – clearer for all readers.

Line numbers on the PDF are cut so it's not possible unfortunately to always refer to the correct line. Sorry for this, we are not sure why it appeared that way, as the manuscript did have line numbers. We will try to be sure this doesn't happen again.

>> Given the surprising result that NGmyb28 plants had lower bacterial loads than NG2 plants, it would be worth also testing NGmyb28 extract media vs. NG2 extract media to see if this is the case here as well, and if similar microbes are enriched. It does not seem this experiment has been done.

We performed bacterial growth assays in leaf extract medium similar to what was done in Fig. 2A. In this assay, we find bacteria to grow mostly to similar levels in NGmyb28 compared to NG2 leaf extract medium. Only one strain, *Xanthomonas campestris* 4E10, grew significantly more in the wild-type NG2, but the growth difference was very small. All other strains either grew less in the NG2 wt extract, or grew similarly in NG2 and NGmyb28 leaf extract medium. Thus, we don't find evidence that breakdown products recruit the same bacteria, further supporting that the intact GLS is the major component contributing this role. We added this data in Fig. S14 and included a description in line 302 and updated the methods accordingly.

In Figure 2A, extract media for NG2, myb28/29, myb28, and Col-0 are shown; it is not clear what the parent genotype of the mutants is, but it seems to be Col-0 from the color scheme. This should also be made more clear.

Yes, this was not very clear. We added boxes around the legends to visually separate NG2 from Col-0 and its two mutants. In addition, we now mention the parent genotype not only in the main text but also in the description of Fig. 2A.

>>The other ruderal plants are never defined. There is also not a picture of them. In the supplementary material they are also incorrectly called “random”. Can the authors provide any more information on what these plants were, or even just some of the plants that this included? Many times other small Brassicaceae can be found among *A. thaliana*. I suppose the authors purposely excluded these, so as to avoid glucosinolates?

This is a good point. The “other” plants were diverse, and we did not photograph or identify all of them by hand. However, we now used chloroplast 16S sequences to identify the plants to the order level. Although chloroplast databases are generally limited, order-level assignment should be conservative so that we now removed the Brassicales found in the “other” plants. We have described this in the methods line 691 and in more detail in the supplementary methods, where also the most common orders identified are listed (most common are Asterales and Poales).

However, removing Brassicales from “other plants” changed the results minimally. This could be because the removed samples were a minority and/or because GLS in those plants (mostly *Draba* sp.) behave differently (for example because those chemotypes don’t recruit bacteria). We can’t know for sure without further study.

>>It’s unclear how the leaf passages are performed. A diagram or more step by step guide would help. Was the media shaken / agitated? The text says intervals were adjusted for growth rate differences. “It took seven days for”... it took 7 days for what? Sufficient cloudiness? How cloudy was sufficiently cloudy? In one part of the text it says “passages on 4MSOB-GLS medium”, and refers to cell density, which doesn’t make sense in the context of a plate. The “on” suggests a plate. The text should be clear always in referring to liquid media or solid media. “plates” and “tubes” would also be acceptable ways of making the distinction between media types.

We used liquid M9 minimal medium supplemented with a single carbon source and performed the first passage after the tubes became visibly turbid. We did not set a particular OD cutoff. We did measure CFUs after the final passage. We have more carefully specified this in the results text (starting at L228) and in the figure description and a detailed description is in the methods starting at line 560. We additionally depicted tubes in an updated version of Fig. 5 for clarity.

>>Fig 3 C... these are differentially abundant taxa according to the legend, but the p values of Wilcoxon tests for some of the comparisons are in the range that would not be considered significant (eg. 0.18). If the initial call of significance is based on a different test, why was the Wilcoxon test also done and which test is better?

DESeq2 analysis is used as a standard tool in microbial ecology to identify differentially abundant taxa between two treatments. That was initially used to identify taxa that may be differentially enriched. However, different approaches often have different sensitivities and it is generally suggested to use multiple analyses for comparison (see for example the NComms analysis here: <https://doi.org/10.1038/s41467-022-28034-z>). Therefore, we as a standard in our group also use the conservative non-parametric Wilcoxon test on the DESeq results to aid interpretation. The reviewer is correct that we didn’t explain this clearly. Thus, we have clarified this in the text L185 and the figure header.

>>Fig 3C... leaves of Arabidopsis often have Methylobacteria, Pseudomonas, Rhizobia, Sphingomonas, and others... but the microbes listed here are generally of lesser abundance. Yet, these apparently are the microbes driving the lower microbial load in the NGmyb28 mutant? Perhaps it's worth comparing the overall microbial community visually for NG2 and for NGmyb28 (eg the stacked column chart like Fig 1B, but showing bacterial families instead of GLS species).

We added stacked bar charts with data agglomerated on family level in Fig. S7 to compare Col-0 and NG2 with their respective aliphatic-GLS-free mutants. We also analyzed the data at higher taxonomic levels in Fig S6. The results confirm our DESeq results and visualize changes in the two families Methylophilaceae and Comamonadaceae which are consistent across the five replicates of NG2 compared to NGmyb28. We think this is the key point, that those taxa identified by DESeq are the ones that are *always* enriched, not necessarily the most abundant ones. Note that there are also higher-abundance taxa included, like *Flavobacterium*. Additionally, taxa like *Pseudomonas* are also highly abundant in some replicates (Fig S7). Given that the NG2 samples had higher loads, these taxa seem to be co-enriched with the more consistent ones that DESeq identified, but the enrichment is apparently not as specific. We tried to clarify this at L193.

>>fig 4B.. it's odd and difficult to track, having the genera indicated by the point color. Perhaps the genera could be indicated by the color of the box at the top of each chart containing the genus name. Also, some of the names are cut by the edges of the box...

We agree this figure was very colorful and difficult to read. To improve this, we colored the boxplots according to the family which is also shown in the legend. The box on top of each chart shows the full genus name. We further improved the readability by removing jittered dots and the red dots which depicted the mean value. Now we only show the median value as part of each boxplot. Overall, we think this makes the figure easier to read without the loss of too much information.

>>Fig 5 is fairly difficult to see in terms of font sizes. Why was there only a single technical replicate of 4MSOB-GLS? Why does the experiment with glucose followed by allyl-GLS say n=3 but shows only 2 technical replicates?

We increased the font size in Fig. 5. The purpose of this experiment was originally to isolate GLS-utilizing bacterial strains to study their dynamics. We had a limited amount of 4MSOB-GLS available (this is mentioned in methods), hence only one replicate. Because of lack of replication in 4MSOB, we have focused on the replicated enrichment in allyl-GLS, including now showing the results of 16S analysis of the individual passages (see new Fig. S11). This shows that a distinct community forms on allyl-GLS even after growth on glucose in the first passage. The reason for only showing two replicates with glucose followed by allyl-GLS is low number of reads in this samples (<100), which we considered to be not reliable and thus we excluded it from the analysis. We kept the single 4MSOB replicate in the figure since it shows that this community is different, but the finding that the allyl-GLS communities are specific can be established just by studying those enrichments (also in comparison with glucose enrichments). We have tried to improve how we talk about this to avoid misunderstanding, starting at L247.

>>The taxa recovered by amplicon sequencing are said to be similar to the taxa growing with 4MSOB-GLS and allyl-GLS media, and Tab S3 and Fig. 5A are mentioned. It would be helpful to see the information about which microbes could be cultured from each media also somehow in the same figure 5A.

We agree that this improves readability and added the information in an updated version of Fig. 5A.

>>”Burkholderiales tend to positively influence community growth”... what does this mean? What data supports this? I don’t recall an experiment in which the addition of Burkholderiales led to the growth of multiple other bacteria.. or am I missing something? I see the point that they are around as long as R3 or another allyl-GLS metabolizer is there. Is this based on literature reports of them being “friendly”? If so, that has not yet been shown for the isolates tested in the study, so probably should not be stated so boldly.

We agree the positive effects on growth were limited, and it was a limited set of taxa. In response to this comment and one comment above from another reviewer, we scrapped the original experiment and started over. In the new setup, we used the experiment only to test whether and how multiple taxa can become enriched with *Rahnella* R3 on allyl-GLS. Specifically, we tested combinations of *Rahnella*, *Pseudomonas*, *Stenotrophomonas* and *Janthinobacterium* in culture on allyl-GLS as C-source and confirm that *Rahnella* is required for growth. In a longer-term three-passage experiment with combinations of *Rahnella*, *Pseudomonas*, and *Janthinobacterium*, we used amplicon sequencing to check their persistence after three passages and untargeted and targeted metabolomics to determine how they affected the metabolome compared to *Rahnella* alone. Figure 7 now focuses on these questions, showing that a) *Rahnella* is required for community growth, b) *Pseudomonas* grows quickly on the spent medium of *Rahnella*, c) *Pseudomonas* and *Janthinobacterium* are maintained through the passages in abundances similar to those in the enrichments from wild leaves, and d) Metabolomics suggests that *Pseudomonas* depletes a number of metabolites associated with *Rahnella*, and *Janthinobacterium* addition tends to further decrease these metabolites. This is strong evidence that these bacteria can persist by cross-feeding from *Rahnella*. As a bonus, we also observed a decrease in the residual allyl-ITC in more diverse cultures, at levels that are likely to be relevant for toxicity and physiological effects. Thus, multiple bacteria appear to both feed from *Rahnella* and potentially benefit the community as a whole. But we have tried to now not focus on Burkholderiales as a whole (also elsewhere in the text). Please refer to Figure 7, Supp Figs 12 and 13 and the completely revised accompanying text with Figure 7. The new experiments have been described in detail in the methods. We also revised the discussion extensively. We do still discuss Burkholderiales, since they are so consistent in our results, but we use solid literature to ground the discussion and to not over-extend our hypotheses (see starting at L403).

>>This sentence is unclear “Assuming this enrichment is indeed due to GLS metabolism, it is possible that bacteria that can access GLSs do not need to be present in high abundances so that we did not detect Yersiniaceae”.

We addressed this as part of a comment of reviewer #1 as well. We worked on the paragraph in the discussion extensively to make our message more clear (starting at L380).

REVIEWERS' COMMENTS

Reviewer #1 (Remarks to the Author):

The authors have dramatically improved the manuscript and addressed all of my concerns, except that the numbering for Supplementary Figures at the end of the Results was a bit weird, especially concerning S12-14. Please have a closer look and fix it.

Reviewer #2 (Remarks to the Author):

The authors replied to all my comments. They performed requested additional experiments (bacterial growth in media containing glucosinolate hydrolysis products) and included the results in the manuscript. The authors also corrected/modified manuscript text and figures accordingly with my comments. I do not see any additional issues with the current version of the manuscript.

Reviewer #3 (Remarks to the Author):

I was overall positive about this work before revisions. I have read the rebuttal document and the revised manuscript. The authors have taken the questions seriously from all reviewers and further invested thought and experiments into improving this manuscript. I am satisfied in particular with the authors' responses to the concerns I raised. There are still open questions but this makes an intriguing and serious contribution to the literature and I have no objections to proceeding to publication.

Reviewer #3 (Remarks on code availability):

I checked that the code is there and looks complete. I did not try to run any of it. The novelty of the manuscript is not its code - the code is for basic statistics and figures, and deeper evaluation would be excessive work for this review.

We thank the reviewers for their comments throughout the process and have responded below to the one remaining request from Reviewer #1.

Please note that in the process of revising Figure 4 based on the Author's checklist, a mistake in the filtering step in the data processing code was noticed. Correcting this showed that a few more taxa were enriched in *A. thaliana* in the wild populations than we had previously reported (See Supplementary Figure 8). However, the patterns were identical to what we first reported and this does not change any interpretation in any way, besides adding further support to the argument. We adjusted the text L214-224 as follows:

Original text:

“*A. thaliana* in populations PB and SW1, both with a 3OHP-GLS chemotype, also enriched similar taxa (Fig. 1B, Fig. S8). At higher taxonomic levels, only *Methylophilaceae* and *Comamonadaceae* were enriched at the NG2 location. Thus, specific genera within a few Burkholderiales families are strongly correlated with allyl-GLS in NG2 and some overlapping genera within the same families are also robustly enriched in *A. thaliana* compared to other plants that do not produce GLS in the wild. Together with other enriched taxa, this strongly supports that GLS supports recruitment of specific bacteria to the leaves of *A. thaliana*. It is worth noting that no taxa associated with aliphatic GLSs or NG2 were enriched in *A. thaliana* in the Woe population, even though these plants share a similar GLS chemotype as NG2 (Fig. S2, Fig. S8). Thus, other genotypic or ecological factors may shape recruitment locally.”

New text:

“*A. thaliana* in populations PB and SW1, both with a 3OHP-GLS chemotype, as well as Woe, another allyl-GLS chemotype, also enriched *Comamonadaceae* genera, but overall, fewer taxa than NG2 (Fig. 1B, Supplementary Fig. 8). At higher taxonomic levels, only *Methylophilaceae* and *Comamonadaceae* were enriched at the NG2 location. Thus, specific genera within a few Burkholderiales families are strongly correlated with allyl-GLS in NG2 and some overlapping genera within the same families are also robustly enriched in *A. thaliana* compared to other plants that do not produce GLS in the wild. Together with other enriched taxa, this strongly supports that GLS supports recruitment of specific bacteria to the leaves of *A. thaliana*. Differences between NG2 and Woe, with the same GLS chemotype (Supplementary Figs. 2 and 8) may suggest that other genotypic or ecological factors shape recruitment locally.

REVIEWERS' COMMENTS

Reviewer #1 (Remarks to the Author):

The authors have dramatically improved the manuscript and addressed all of my concerns, except that the numbering for Supplementary Figures at the end of the Results was a bit weird, especially concerning S12-14. Please have a closer look and fix it.

We fixed this by changing the order of these supplementary figures, which we had left out of order.

Reviewer #2 (Remarks to the Author):

The authors replied to all my comments. They performed requested additional experiments (bacterial growth in media containing glucosinolate hydrolysis products) and included the results in the manuscript. The authors also corrected/modified manuscript text and figures accordingly with my comments. I do not see any additional issues with the current version of the manuscript.

Reviewer #3 (Remarks to the Author):

I was overall positive about this work before revisions. I have read the rebuttal document and the revised manuscript. The authors have taken the questions seriously from all reviewers and further invested thought and experiments into improving this manuscript. I am satisfied in particular with the authors' responses to the concerns I raised. There are still open questions but this makes an intriguing and serious contribution to the literature and I have no objections to proceeding to publication.

Reviewer #3 (Remarks on code availability):

I checked that the code is there and looks complete. I did not try to run any of it. The novelty of the manuscript is not its code - the code is for basic statistics and figures, and deeper evaluation would be excessive work for this review.